# Concept Component Analysis: A Principled Approach for Concept Extraction in LLMs

## Abstract

Developing human understandable interpretation of large language models (LLMs) becomes increasingly critical for their deployment in essential domains. Mechanistic interpretability seeks to mitigate the issues through extracts human-interpretable process and concepts from LLMs' activations. Sparse autoencoders (SAEs) have emerged as a popular approach for extracting interpretable and monosemantic concepts by decomposing the LLM internal representations into a dictionary. Despite their empirical progress, SAEs suffer from a fundamental theoretical ambiguity: the well-defined correspondence between LLM representations and human-interpretable concepts remains unclear. This lack of theoretical grounding gives rise to several methodological challenges, including difficulties in principled method design and evaluation criteria. In this work, we show that, under mild assumptions, LLM representations can be approximated as a linear mixture of the log-posteriors over concepts given the input context, through the lens of a latent variable model where concepts are treated as latent variables. This motivates a principled framework for concept extraction, namely Concept Component Analysis (ConCA), which aims to recover the log-posterior of each concept from LLM representations through a unsupervised linear unmixing process. We explore a specific variant, termed sparse ConCA, which leverages a sparsity prior to address the inherent ill-posedness of the unmixing problem. We implement 12 sparse ConCA variants and demonstrate their ability to extract meaningful concepts across multiple LLMs, offering theory-backed advantages over SAEs—namely, a clear unmixing target, principled sparsity placement, and improved alignment with latent concepts.

## 1 Introduction

One of the critical questions surrounding the practical application of LLMs is the extent to which and how the concepts they espouse are ground in reality. The more general question is whether a model trained only on natural language can develop representations of concepts grounded in the real world (Bowman, 2024; Naveed et al., 2023). Understanding this relationship is crucial, as it informs not only how we interpret model mechanism, but also how we can systematically analyze, evaluate, and manipulate these representations. A promising approach to investigating such questions is to extract meaningful semantic units, i.e., human-interpretable concepts, embedded within the models' internal representations and behaviors (Singh et al., 2024). By studying these units, we can begin to probe which aspects of a model's behavior are aligned with human-interpretable concepts, and how multiple concepts interact to generate model outputs.

### 1.1 Revisiting SAEs for Concept Extraction

**Sparse autoencoders (SAEs).** Recently, SAEs have been explored for this purpose (Elhage et al., 2022; Bricken et al., 2023; Huben et al., 2023), offering a potential perspective through which to analyze model behavior, including how such concepts interact and compose to generate the next token. Technically, SAEs learn a set of features whose *linear* combinations can reconstruct the internal representations of LLMs, while enforcing a *sparsity* prior on the features, in the hope that each feature corresponds to a monosemantic concept (Huben et al., 2023; Gao et al., 2025; Braun

Figure 1: We introduce a latent variable generative model in which observed the input context $\mathbf{x}$ and next token $y$, arises from an unknown underlying process over latent concepts $\mathbf{z}$ (Sec. 2.1). Under this model, we show that LLM representations $\mathbf{f_x(x)}$, learned by next-token prediction, can be approximated as a linear mixture of the column vector obtained by stacking log-posteriors of individual latent concepts $\log p(z_i|\mathbf{x})$, conditioned on the input, i.e., $\mathbf{f_x(x)} \approx \mathbf{A}\big[[\log p(z_1 \mid \mathbf{x})]_{z_1}; \dots; [\log p(z_\ell \mid \mathbf{x})]_{z_\ell}\big] + \mathbf{b}$, where $\mathbf{A}$ is a mixing matrix and $\mathbf{b}$ is a constant (Sec. 2.2). Motivated by this, we propose Concept Component Analysis (ConCA), a method for linearly unmixing LLM representations $\mathbf{f_x(x)}$ to recover the log-posteriors over individual latent concepts $\log p(z_i|\mathbf{x})$ (Sec. 3).

et al., 2024; Rajamanoharan et al., 2024a;b; Mudide et al., 2024; Chanin et al., 2024; Lieberum et al., 2024; He et al., 2024; Karvonen et al., 2024; Bussmann et al., 2024).

**Hypotheses Behind SAEs.** Linearity and sparsity, the two key components of SAEs, are jointly expected to promote the emergence of monosemantic and interpretable concepts. The justification for these two components primarily relies on two key hypotheses, (i) the linear representation hypothesis and (ii) the superposition hypothesis. The former suggests that concepts are often encoded linearly in LLMs (Tigges et al., 2023; Nanda et al., 2023; Moschella et al., 2022; Park et al., 2023; Li et al., 2024; Gurnee et al., 2023; Rajendran et al., 2024; Jiang et al., 2024), enabling them to be recovered via linear decoding. The latter argues that LLMs tend to represent more features than they have neurons for, leading to overlapping (i.e., superimposed) representations within the same neurons (Elhage et al., 2022). To make such representations reliable and interpretable, features should activate sparsely, reducing interference between them (Elhage et al., 2022; Huben et al., 2023).

## 1.2 MOTIVATION AND CONTRIBUTIONS

While these two hypotheses support SAEs, the deeper theoretical question remains unresolved.

> ⚑ **Key Problem.** What is the theoretical relationship between LLM representations and human-interpretable concepts?

**A Deeper Look into SAEs.** Without a clear answer, both principled method design and evaluation become major concerns. In terms of method design, for example, while the decoder in SAEs reconstructs representations through *linear* combinations of learned features, the encoder typically includes a *nonlinear* activation function, typically `Relu`, following a linear layer. This asymmetry raises a concern about the rationale for introducing the nonlinear activation functions. Moreover, it remains unclear whether sparsity should be imposed directly on the feature space learned by SAEs, or instead on a transformed space derived from it, given the unclear relationship between these features and the underlying concepts. This unclear relationship, on the evaluation side, also makes principled assessment difficult, i.e., it remains unclear what criteria should be used to determine whether a feature meaningfully captures a concept, as also recognized in recent works (Makelov et al., 2024; Gao et al., 2025; Kantamneni et al., 2025).

**Contributions.** We propose a principled approach for extracting concepts from LLM representations, grounded in a well-defined theoretical relationship between the representations and human-interpretable concepts (see Figure 1). We begin by analyzing this relationship through the lens of a latent variable model, in which text data are generated by an unknown process over latent, human-interpretable concepts. We show that, under mild conditions, LLM representations learned by next-token prediction frameworks can be approximately expressed as a linear mixture of the

logarithm of the posteriors of individual latent concepts, conditioned on the input context. Based on this insight, we introduce a principled approach, that we label Concept Component Analysis (ConCA), which aims to invert the linear mixture to recover the log-posterior of each concept in an unsupervised manner. We propose a specific variant of ConCA, referred as Sparse ConCA, which incorporates a sparsity prior to regularize the solution space, motivated by the widespread adoption of the superposition hypothesis. We emphasize that alternative regularization strategies remain flexible and open for future exploration. We evaluate the proposed Sparse ConCA using linear probing with counterfactual text pairs, a theoretically motivated supervised method for concept extraction, and benchmark its performance against SAE variants across multiple model scales and architectures (Pythia (Biderman et al., 2023), Gemma3 (Team et al., 2025), Qwen3 (Team, 2025)). We further test it on a downstream task spanning 113 datasets, empirically confirming the advantages of ConCA.

## 2 What Do Representations in LLMs Learn?

In this section we establish a theoretical connection between LLM representations learned through next-token prediction framework and human-interpretable concepts. To this end, we first construct a latent variable model (LVM) in which observed text data are generated by an unknown process over latent variables representing human-interpretable concepts. We then show that, when LLMs are trained on the observed data using a next-token prediction framework, their learned representations can be approximated as a linear mixture of the log-posteriors of individual latent variables, conditioned on the input context. This perspective not only deepens our understanding of how human-interpretable concepts are organized within LLM representations, but more importantly, it provides a principled foundation for extracting concepts from the representations. We define concept as follows:

**Definition 2.1.** A *concept* is defined as a discrete latent variable

$$z_i \in \mathcal{V}_i, \qquad |\mathcal{V}_i| = k_i,$$

where each value in $\mathcal{V}_i$ corresponds to a distinct, human-interpretable semantic attribute (e.g., tense, plurality, sentiment, syntactic role, or topic). The full latent configuration is given by $\mathbf{z} = (z_1, \dots, z_\ell)$, whose components specify the underlying semantic factors that give rise to the observed input context $\mathbf{x}$ and the next token $y$ through the latent generative process.

### 2.1 Preliminary: A Discrete Latent Variable Generative Model For Text

We begin by a LVM in which human-interpretable concepts are modeled as latent variables governing the generation of text data (Liu et al., 2025a). Formally, both the observed context $\mathbf{x}$ and the next token $y$ are assumed to be generated from a set of latent variables $\mathbf{z}$. Here $\mathbf{x}$ and $y$ serve as input to the next-token prediction objective used to train LLMs. A human-interpretable concept is formally defined as a latent variable $z_i$ that captures a human-interpretable factor underlying the generation of text data, such as a topic, sentiment, syntactic role, or tense. Notably, arbitrary interdependencies or structural relationships among the latent variables are allowed. We assume the observed variables $\mathbf{x} \in \mathcal{V}^n$ and $y \in \mathcal{V}$, and the latent variables $\mathbf{z} = (z_1, \dots, z_\ell) \in \mathcal{V}_1 \times \dots \times \mathcal{V}_\ell$ to be discrete[1], with $z_i \in \mathcal{V}_i, |\mathcal{V}_i| = k_i, i = 1, \dots, \ell$. Under this formulation, the joint distribution over the observed context $\mathbf{x}$ and next token $y$ is given by:

$$p(\mathbf{x}, y) = \sum_{\mathbf{z}} p(\mathbf{x}|\mathbf{z}) \, p(y|\mathbf{z}) \, p(\mathbf{z}) \,, \tag{1}$$

where $p(\mathbf{z})$ is a prior over the latent concepts, and $p(\mathbf{x}|\mathbf{z})$ and $p(y|\mathbf{z})$ model the conditional generation of context and next token, respectively.

### 2.2 Representations in LLMs Linearly Encode Log-Posteriors over Concepts

Intuitively, since the latent concepts $\mathbf{z}$ characterize the underlying generative factors of the text data, as defined in Eq. 1, the representations learned from such data should encode information about these concepts. To examine in detail how these representations capture latent concepts, we now turn to the next-token prediction framework, which serves as the standard training framework for LLMs.

---

[1] A detailed justification for the discrete assumption can be found in Liu et al. (2025a).

Specifically, the next-token prediction framework models the conditional distribution of the next token $y$ given the input context $\mathbf{x}$[2], as follows:

$$p(y|\mathbf{x}) = \frac{\exp\left(\mathbf{f}(\mathbf{x})^T \mathbf{g}(y)\right)}{\sum_{y_i} \exp\left(\mathbf{f}(\mathbf{x})^T \mathbf{g}(y_i)\right)}. \tag{2}$$

Here, $y_i$ denotes a specific value of the output token $y$, $\mathbf{f}(\mathbf{x}) \in \mathbb{R}^m$ maps the input $\mathbf{x}$ into a $m$-dimensional (depending on the specific model used) representation space, and $\mathbf{g}(y) \in \mathbb{R}^m$ retrieves the classifier weight vector corresponding to token $y$, i.e., the look-up table used for prediction.

Given the generative model from Eq. 1 and the inference model from Eq. 2, our goal is to formally characterize how the learned representations $\mathbf{f}(\mathbf{x})$ relate to the latent concepts $\mathbf{z}$. In particular, we seek to establish a precise mathematical relationship, thereby serving as the theoretical foundation for concept component analysis developed in Sec. 3. We now present the following key result:

**Theorem 2.2.** *Suppose latent variables $\mathbf{z}$ and the observed variables $\mathbf{x}$ and $y$ follow the generative models defined in Eq. 1. Assume the following holds:*

*(i) **(Diversity Condition)** There exist $m + 1$ values of $y$, such that the matrix $\mathbf{L} = \big(\mathbf{g}(y = y_1) - \mathbf{g}(y = y_0), ..., \mathbf{g}(y = y_m) - \mathbf{g}(y = y_0)\big)$ of size $m \times m$ is invertible,*

*(ii) **(Informational Sufficiency Condition)** The conditional entropy of the latent concepts given the context is close to zero, i.e., $H(\mathbf{z}|\mathbf{x}) \to 0$,*

*then the representations $\mathbf{f}(\mathbf{x})$ in LLMs, which are learned through the next-token prediction framework, are related to the true latent variables $\mathbf{z}$, by the following relationship:*

$$\mathbf{f}(\mathbf{x}) \approx \mathbf{A}\big[[\log p(z_1 \mid \mathbf{x})]_{z_1}; \ldots; [\log p(z_\ell \mid \mathbf{x})]_{z_\ell}\big] + \mathbf{b}, \tag{3}$$

*where $\mathbf{A}$ is a $m \times (\sum_{i=1}^{\ell} k_i)$ matrix, and $\mathbf{b}$ is a bias vector[3].*

**Justification for Conditions (i) and (ii).** Condition (i) is closely related to the data diversity assumption in earlier work for identifiability analysis in the context of nonlinear independent component analysis (Hyvarinen & Morioka, 2016; Hyvarinen et al., 2019; Khemakhem et al., 2020), and has recently been employed to identifiability analyses of latent variables in the context of LLMs (Roeder et al., 2021; Marconato et al., 2025; Liu et al., 2025a). Intuitively, Condition (i) implies that there is a sufficiently large number of distinct values of $y$ that the $m$ difference vectors $\mathbf{g}(y_i) - \mathbf{g}(y_0)$ (for $i = 1, \ldots, m$) span the image of $\mathbf{g}$. This is a mild assumption, as pointed out by Roeder et al. who emphasized that the set of $m + 1$ values $\{y_i\}_{i=0}^m$ required to generate difference vectors $\mathbf{g}(y_i) - \mathbf{g}(y_0)$ that are linearly dependent has measure zero, given that both the initialization and subsequent updates of the parameters of $\mathbf{g}$ are stochastic. Turning to Condition (ii), it can be seen as a *mild relaxation* of a standard invertibility assumption commonly adopted for identifiability analysis within causal representation learning community, i.e., the mapping from $\mathbf{z}$ to $\mathbf{x}$ in the generative model (Eq. 1) is assumed to be deterministic and invertible. This implies $H(\mathbf{z} \mid \mathbf{x}) = 0$, to ensure exact recovery of the latent variables. By contrast, our assumption only requires that $H(\mathbf{z} \mid \mathbf{x}) \to 0$, allowing for approximate invertibility from $\mathbf{z}$ to $\mathbf{x}$ in practice. This relaxation also implies that only an approximate recovery is achievable, as shown in Eq. 3.

💡 **Inspiration:** Eq. 3 in Theorem 2.2 implies that LLM representations, learned through the next-token prediction framework, are essentially a linear mixture of the log-posteriors of individual latent concepts given the input context, that is, the $\log p(z_i|\mathbf{x})$. This provides a theoretical foundation for exploring individual concepts by linearly unmixing the representations, which motivates the development of our *ConCA* framework in Sec. 3.

---

[2]More rigorously, this assumes a parametric form of the conditional distribution $p(y|\mathbf{x})$ as a softmax over inner products in an optimally discriminative representation space. Such optimality assumption is canonical for establishing clear and meaningful identifiability results (Hyvarinen & Morioka, 2016; Hyvarinen et al., 2019; Khemakhem et al., 2020).

[3]Here, $[\log p(z_i \mid \mathbf{x})]_{z_i}$ denotes the column vector of log-probabilities of all possible values of $z_i$. $\big[[\log p(z_1 \mid \mathbf{x})]_{z_1}; \ldots; [\log p(z_\ell \mid \mathbf{x})]_{z_\ell}\big]$ represents the column vector obtained by stacking these vectors.

# 3 ConCA: A Principled Approach for Concept Extraction in LLMs

Grounded in Theorem 2.2, we now introduce *ConCA*, a principled approach that recovers the *log posteriors* of individual latent concepts conditional on the input context, i.e., $\log p(z_i|\mathbf{x})$, by inverting the linear mixture in Eq. 3, in a unsupervised way. This enables us to decompose LLM representations $\mathbf{f}(\mathbf{x})$ into interpretable concept-level components.

## 3.1 Challenges in Designing ConCA

Recovering $\log p(z_i|\mathbf{x})$ from $\mathbf{f}(\mathbf{x})$ presents two main challenges:

> ⚠ **Key Challenges:** ① **Ill-posed inverse problem:** The inversion of Eq. 3 is inherently ill-posed because there exist many possible solutions that produce the same LLM representations. Without additional constraints or regularization, the solution space is large and ambiguous, leading to non-unique decompositions. ② **Underdetermined problem and overfitting:** The issue above is particularly critical when the dimension of the latent concept space exceeds that of the observed representations (i.e., $\ell > m$), corresponding to the well-known underdetermined case. This occurs because, from the model's perspective, the number of degrees of freedom to be estimated increases, which not only significantly expands the set of possible solutions but also amplifies the risk of overfitting.

To address challenge ①, we impose a sparsity prior, requiring that for each context $\mathbf{x}$, only a small subset of latent concepts $\mathbf{z}$ is activated. We refer to this variant as *sparse ConCA*. This inductive bias is motivated by two considerations: (i) it aligns with the superposition hypothesis, a widely discussed phenomenon in the study of SAEs, and (ii) sparsity ensures the identifiability of latent factors under certain assumptions, i.e., the individual log-probabilities of latent concepts can be uniquely recovered, as established in the theory of sparse dictionary learning (Elad & Bruckstein, 2002; Gribonval & Schnass, 2010; Spielman et al., 2012; Arora et al., 2014). Importantly, we highlight the following:

> ★ **Highlight:** Unlike SAEs, which recover concepts loosely, like a blurry sketch, sparse ConCA recovers a clearly defined concepts, i.e., the log-posteriors $\log p(z_i|\mathbf{x})$.

This key distinction fundamentally changes how sparsity should be interpreted and enforced. In SAEs and sparse coding, sparsity is directly imposed on the latent feature space, where values near zero correspond to inactive features. In ConCA, however, the situation is inverted. Specifically, the latent feature space in ConCA corresponds to the log-posteriors, i.e., $\log p(z_i|\mathbf{x})$. A value of $\log p(z_i|\mathbf{x}) = 0$ corresponds to $p(z_i|\mathbf{x}) = 1$, meaning the concept $z_i$ is *fully active* rather than inactive. Consequently, sparsity in ConCA should be enforced in the exponential form of the latent feature space, i.e., the posteriors $p(z_i|\mathbf{x})$, ensuring that only a small subset of concepts is truly active.

Despite its advantages, sparsity alone does not fully resolve the challenge of overfitting, particularly in underdetermined settings where the dimensionality of $\mathbf{z}$ is much higher than that of $\mathbf{f}$, as mentioned in challenge ②. While the sparsity prior helps restrict the solution space, it does not guarantee generalizable or semantically meaningful decompositions in practical scenarios. Although theoretical results provide identifiability guarantees under ideal conditions, real-world challenges, such as limited data and optimization difficulties, often violate these conditions, leading to potential overfitting and less reliable concept recovery. In such settings, multiple sparse solutions may fit the observed representation equally well, and some may capture meaningless noise or non-semantic patterns rather than true underlying concepts. Therefore, techniques to mitigate overfitting may be both useful and even necessary in real applications.

## 3.2 Sparse ConCA: Architecture and Training Objective

According to the analysis above, we propose sparse ConCA as follows:

$$\hat{\mathbf{z}} = \mathcal{R}(\mathbf{W}_e\mathbf{f}(\mathbf{x}) + \mathbf{b}_e), \qquad \hat{\mathbf{f}}(\mathbf{x}) = \mathbf{W}_d\hat{\mathbf{z}} + \mathbf{b}_d. \qquad (4)$$

This is a typical autoencoder architecture, where $\mathcal{R}(\cdot)$ denotes a general regularization module applied to mitigate overfitting, including but not limited to Dropout, and LayerNorm, as the analysis above to

| Aspect | SAEs | ConCA |
|---|---|---|
| **Theoretical grounding** | (i) Linear representation hypothesis (ii) Superposition hypothesis | Theorem 2.2 |
| **Objective** | Recover monosemantic features | Recover $\log p(z_i|\mathbf{x})$ |
| **Architecture** | Encoder: linear + nonlinear activation Decoder: linear | Encoder: linear + module for overfitting Decoder: linear |
| **Role of sparsity** | On feature space | On exp-transformed feature space |
| **Evaluation criterion** | Heuristic, lacks principled metric | Theoretically motivated (Sec. 4) |

Table 1: Comparison between SAEs and the proposed ConCA. ConCA provides a principled, theoretically grounded framework for disentangling LLM representations, while SAEs are largely motivated by empirical hypotheses.

address the challenge ②. $\mathbf{W}_e$ and $\mathbf{W}_d$ are learnable weight matrices of the encoder and decoder, respectively, and $\mathbf{b}_e$, $\mathbf{b}_d$ are the corresponding biases. The vector $\hat{\mathbf{z}}$ corresponds to an estimate of $[\log p(z_i \mid \mathbf{x})]_{z_i}$. Let the set of all learnable parameters be $\Theta = \{\mathbf{W}_e, \mathbf{b}_e, \mathbf{W}_d, \mathbf{b}_d\}$. We train the proposed sparse ConCA by minimizing the following objective with respect to $\Theta$:

$$\min_{\Theta} \quad \mathbb{E}_{\mathbf{x}} \left[ ||\hat{\mathbf{f}}(\mathbf{x}) - \mathbf{f}(\mathbf{x})||_2^2 + \alpha \mathcal{S}(\mathbf{g}(\hat{\mathbf{z}})) \right], \tag{5}$$

where we apply $\mathbf{g}(\cdot)$ to the representations $\hat{\mathbf{z}}$ (corresponding to log-posterior in theory), to map them back into the probability domain, where sparse activation patterns can be meaningfully enforced, as motivated by the analysis above to address challenge ①. Ideally, the exact $\exp(\cdot)$ function would be optimal, but it is prone to numerical instability and exploding gradients when $\hat{\mathbf{z}}$ takes large values. Therefore, we employ a smooth surrogate in practice, see Sec. 4 for further implementation details. This regularization function $\mathcal{S}(\cdot)$ is then applied so as to encourage sparsity on $\mathbf{g}(\hat{\mathbf{z}})$. This can be implemented using standard sparsity constraints such as $L_1$ regularization or structured sparsity variants [4]. The hyperparameter $\alpha$ controls the trade-off between reconstruction fidelity and sparsity, allowing the model to be tuned to the expected degree of sparsity.

The key distinctions between our proposed ConCA framework and SAEs are summarized in Table 1.

## 4 EXPERIMENTS

We train the proposed sparse ConCA on a subset of the Pile (the first 200 million tokens) (Gao et al., 2020). The regularization function $\mathcal{R}(\cdot)$ is implemented using 4 normalization strategies, including `LayerNorm` (Ba et al., 2016), `Dropout` (Srivastava et al., 2014), `BatchNorm` (Ioffe & Szegedy, 2015), and `GroupNorm` (Wu & He, 2018). For the function $\mathbf{g}(\cdot)$, not exponential function directly, we explore the exponential with 3 different activation functions, `SELU` (Klambauer et al., 2017), `SoftPlus` (Dugas et al., 2000), and `ELU` (Clevert et al., 2015). Although they are not exact exponentials, these functions preserve exponential-like behavior for small (i.e., negative) values, ensure numerical and gradient stability, and provide smooth surrogates suitable for applying sparsity regularization. In total, we implement 12 sparse ConCA variants across these configurations. Sparsity, i.e., $\mathcal{S}(\cdot)$, is primarily enforced via $L_1$ normalization in this work, other choices remain flexible. To evaluate the effect of model scale, we use representations from Pythia models of varying sizes: 70M, 1.4B, and 2.8B Biderman et al. (2023). To assess model generalization, we also test across different architectures, including Pythia-1.4B, Gemma3-1b (Team et al., 2025), and Qwen3-1.7B (Team, 2025). We compare the proposed sparse ConCA with various SAE variants, including top-$k$ SAE (Gao et al., 2025), batch-top-$k$ SAE (Bussmann et al., 2024), $p$-annealing SAE (Karvonen et al., 2024).

We evaluate sparse ConCA using two metrics designed to assess both faithfulness and interpretability:

- **Reconstruction loss** captures how well the original LLM representations are preserved after decomposition. Since our goal is to reveal the internal structure of the model without altering its

---

[4]We emphasize that sparsity is a design choice, other forms of regularization are potentially applicable. For instance, non-negativity (Lee & Seung, 1999; Hoyer, 2004) or bounded-range constraints (Cruces, 2010; Erdogan, 2013), given that the learned features are expected to correspond to probabilities.

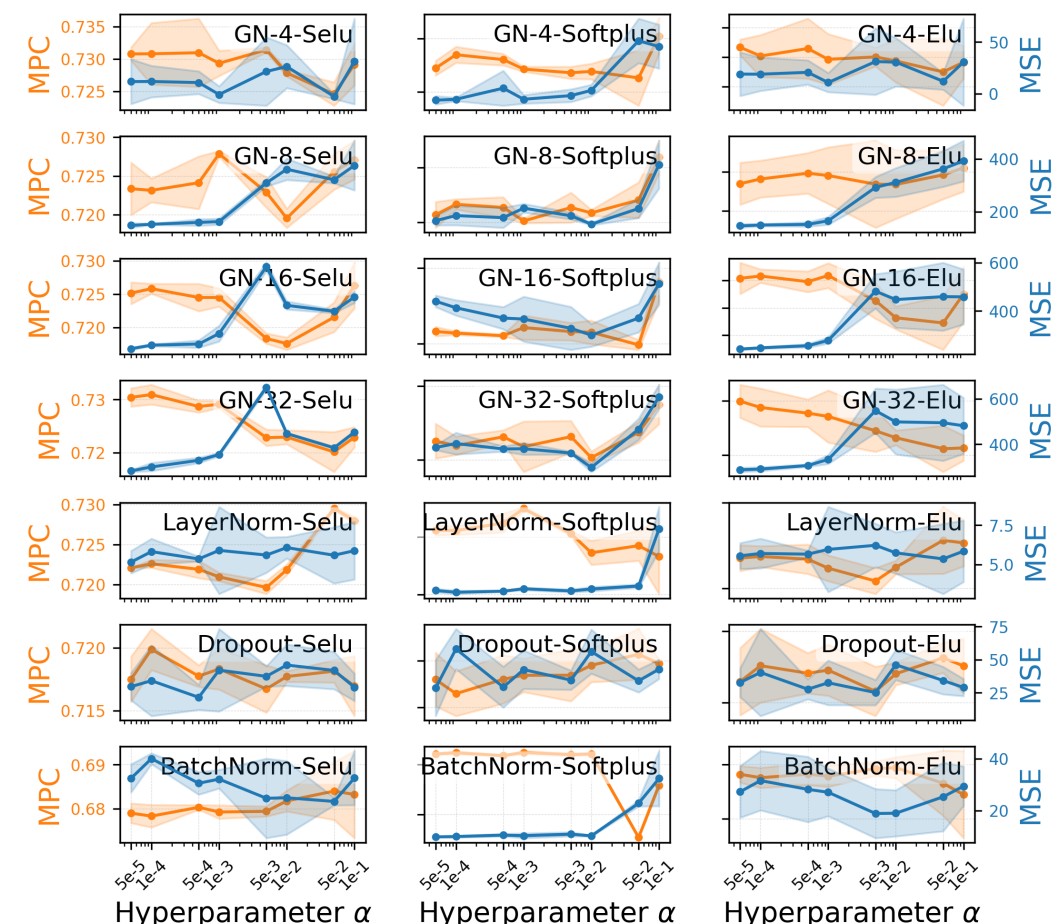

Figure 2: Ablation study of 4 different normalization methods and 3 activation functions. For `GroupNorm`, the number of groups (`num_groups`) is set to 4, 8, 16, or 32. Left axis shows Mean of Pearson Correlation (MPC), right axis shows MSE. Each subplot corresponds to one combination of normalization method and activation function, with each configuration run three times. Across all configurations, ConCA exhibits a remarkably stable correlation regime (MPC $\approx$ 0.72–0.74, excluding BatchNorm), with performance largely insensitive to the sparsity level. Notably, the MSE is affected by the number of groups used in GroupNorm, i.e., fewer groups lead to lower MSE. Overall, LayerNorm emerges as a strong choice, offering consistently good performance in both MPC and MSE.

behavior, low reconstruction loss is essential to ensure that concept extraction introduces minimal distortion. Specifically, we use mean squared error (MSE) as our reconstruction loss metric.

- **Pearson correlation** quantifies how well the ConCA-extracted features align with theoretically consistent supervised estimates of $\log p(z_i|\mathbf{x})$ for each latent concept $z_i$. Specifically, for each latent concept $z_i$, we construct counterfactual pairs that differ only in the value of $z_i$ while keeping all other variables unchanged, and train a linear classifier to predict this difference, yielding a supervised estimate of $\log p(z_i|\mathbf{x})$. This estimator is theoretically motivated, see Sec. F[5]. We then compute the Pearson Correlation (PC) between this supervised estimate of $\log p(z_i|\mathbf{x})$ and the unsupervised ConCA feature. Higher correlation indicates more accurate recovery.

To compute Pearson correlation, we require counterfactual text pairs as mentioned above. However, constructing such counterfactual pairs is highly challenging due to the complexity and subtlety of natural language, as noted in prior works (Park et al., 2023; Jiang et al., 2024), and remains non-trivial even for human annotators. For our evaluation, we adopt 27 counterfactual pairs from Park et al.

---

[5]We note that Liu et al. (2025a) provide a similar approach, but our result is derived from Theorem 2.2.

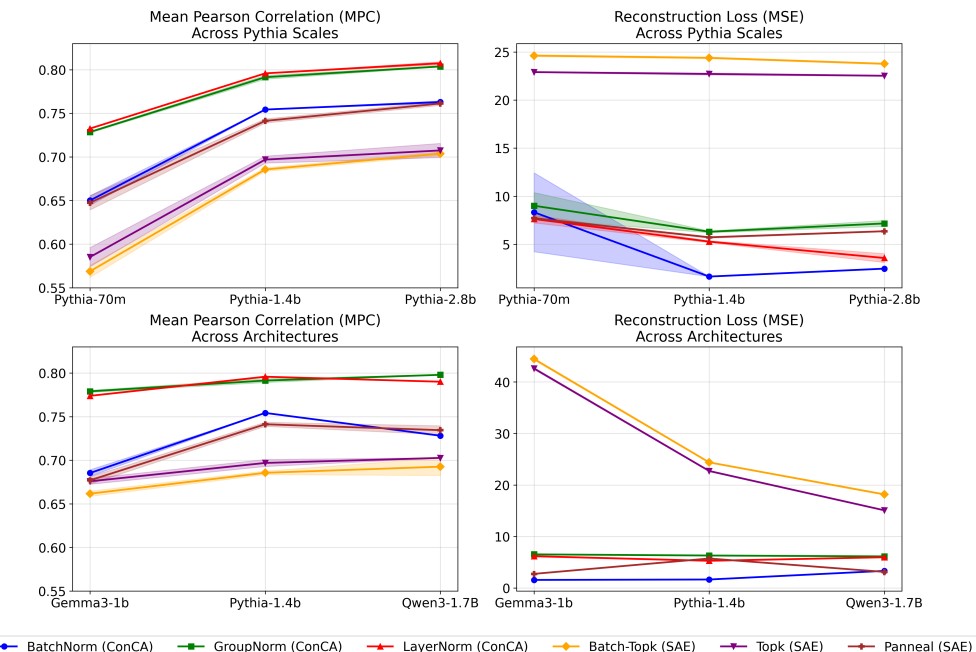

Figure 3: Comparison of SAE variants and the proposed ConCA variant across different scales and architectures. The left two shows the results for Pythia family with varying sizes (70m, 1.4b, 2.8b), while the right compares different architectures across multiple models (Gemma-3-1b, Pythia-1.4b, Qwen3-1.7b). Pearson correlation (left axis) and MSE (right axis) are reported for each method. ConCA variants (BatchNorm, GroupNorm, LayerNorm), overall, achieve higher MPC than SAE baselines (Top-k, Batch-Top-k, Panneal), with LayerNorm-ConCA performing best across all settings (approximately 0.70–0.80). SAE methods remain in a lower band (approximately 0.60–0.70) and show weaker gains with model scale. Reconstruction error (MSE) varies substantially across methods: only Panneal (SAE) obtain lower MSE, whereas ConCA maintains strong both MPC and MSE. Overall, the figure highlights that ConCA more reliably extracts concepts, robust across model size and architecture. Full numerical mean and std values, see Sec. Q.

(2023), each differing in a single concept, as testing dataset. These pairs were derived from the Big Analogy Test dataset (Gladkova et al., 2016).

**Ablation Study** We first conduct an ablation study over normalization strategies, activation functions, and sparsity strength as mentioned above, to understand the design choices of sparse ConCA. In total, this yields 21 configurations (For `GroupNorm`, the number of groups (`num_groups`) is set to 4, 8, 16, and 32, respectively). Each configuration is trained with varying sparsity coefficients $\alpha \in \{1e^{-1}, 5e^{-2}, 1e^{-2}, \ldots, 5e^{-5}\}$, and every experiment is repeated three times with training on Pythia-70M. We report results on the two key evaluation metrics as mentioned, i.e., reconstruction loss and Pearson correlation. Both metrics are summarized in Figure 2, where the left $y$-axis shows correlation and the right $y$-axis shows reconstruction loss.

**Findings.** Three main observations emerge:

- **Normalization.** The primary role of normalization methods in our framework is to mitigate overfitting. For `GroupNorm`, increasing the number of groups (`num_groups`) tends to result in slightly higher reconstruction loss. Conversely, `LayerNorm` achieves the lowest reconstruction loss among the considered methods (`Dropout`, `BatchNorm`). This trend suggests that full-feature normalization, as performed by `LayerNorm`, better preserves the overall structure of LLM representations. This may be because, `LayerNorm` stabilizes per-sample activations and preserves global feature correlations, which helps ConCA recover concept log-posteriors more faithfully and generalize better than the others.
- **Activation.** The purpose of the activation functions is to serve as a surrogate for the exact $\exp(\cdot)$ function, enabling more effective enforcement of sparsity. Roughly, for `GroupNorm`, all three

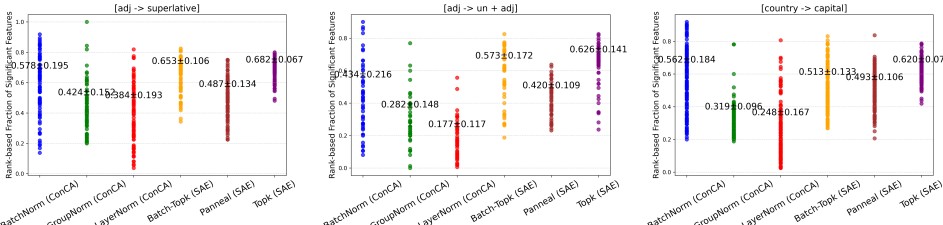

Figure 4: Rank-based fraction of features exhibiting significant changes between counterfactual pairs for SAE and ConCA variants. ConCA shows smaller feature variations, indicating more stable feature responses under counterfactual pairs.

activations (`SELU`, `ELU`, and `SoftPlus`) perform similarly, with Pearson correlation values around 0.725–0.73. In the context of `LayerNorm` and `Dropout`, `SoftPlus` appears slightly better than the other two, whereas for `BatchNorm`, `SELU` seems slightly better.

- **Sparsity.** The sparsity coefficient $\alpha$ controls a trade-off: too large a value may introduce excessive information loss, while too small a value fails to induce meaningful structure. Overall, across the range $[5e^{-3}, \ldots, 1e^{-2}]$, the performance in both reconstruction loss and Pearson correlation remains relatively stable.

**Takeaway.** The ablation study demonstrates that careful choices of normalization and activation functions significantly improve the performance of sparse ConCA. In the following experiments, considering both reconstructuion loss and Pearson Correlation, we focus on the most promising configurations: `GroupNorm` with `num_groups= 4` and `Softplus`, `LayerNorm` with `Softplus`, `BatchNorm` with `Softplus`, labeled as Groupnorm, LayerNorm, and BatchNorm in the following, respectively. For all of these, the sparsity hyperparameter, we set $\alpha = 1e^{-4}$. These design choices are consistent across repeated trials, highlighting the stability of the proposed sparse ConCA.

**Comparison on Counterfactual Pairs.** We next conduct experiments comparing various SAE variants, including including Topk SAE (Gao et al., 2025), Batch-Topk SAE (Bussmann et al., 2024), P-anneal SAE (Karvonen et al., 2024), and the proposed ConCA configurations mentioned in **Takeaway** above. We conduct these experiments across different scales of the Pythia family to evaluate the scalability and effectiveness of each method. Each method is run across multiple random seeds to ensure robustness, and we present both the mean and standard deviation of the metrics. The left in Figure 3 highlights how the proposed ConCA configurations consistently achieve higher Pearson correlation while maintaining competitive reconstruction loss compared to the SAE variants, as model size increases from Pythia-70m to Pythia-2.8b. This performance advantage is mainly due to the theoretical grounding of ConCA. Specifically, ConCA employs a principled framework with sparsity on the exponentiated feature space, whereas SAEs rely on heuristic assumptions, resulting in more interpretable and accurate monosemantic concepts across Pythia scales. Notably, when considering both MSE and Pearson correlation, the LayerNorm configuration emerges as the better choice. The advantages of ConCA are further highlighted across different model families, including Gemma-3-1b, Pythia-1.4b, and Qwen3-1.7B, as shown in the right of Figure 3. ConCA configurations generally outperform SAE variants in both reconstruction and Pearson correlation, demonstrating the robustness and broad applicability of ConCA. Figure 4 shows the rank-based fraction of features that change significantly between counterfactual pairs, indicating that ConCA produces smaller feature variations than SAE variants under counterfactual conditions. See Sec. H for details on how this metric is computed and for additional visualization results.

**Downstream Tasks** In the final stage, we conduct a series of few-shot linear probing experiments to evaluate how well the features extracted by SAEs and ConCA capture monosemantic, human-interpretable concepts. This evaluation is particularly relevant because disentangled representations tend to transfer easily and robustly, making them especially suitable for few-shot learning and out-of-distribution shift tasks (Fumero et al., 2023).

To this end, we collect 113 binary classification datasets from Kantamneni et al. (2025) and use them to train linear classifiers on features extracted by SAEs and ConCA variants under limited training samples, specifically 4, 8, 16, 32, and 128 samples drawn randomly. After training, we evaluate the

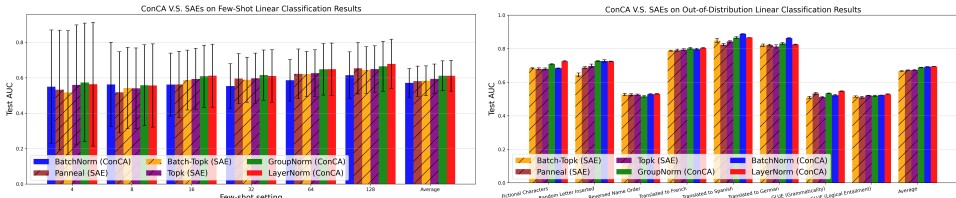

Figure 5: Test AUC of SAE variants and the proposed ConCA variants under different few-shot settings across 113 datasets (left), and out-of-distribution tasks across 8 datasets (right), respectively. An example of visualization can be found in Sec. I.

linear classifiers and report the Area Under the Receiver Operating Characteristic Curve (AUC). The left panel of Figure 5 shows a trend where ConCA often achieves higher AUC than SAE variants in few-shot settings, particularly for LayerNorm and GroupNorm variants, although the differences are not statistically significant under the current sample size.

Furthermore, we extend our evaluation to 8 *out-of-distribution* (OOD) datasets from Kantamneni et al. (2025), which are designed to test robustness under distributional shifts. These datasets include fictional character substitution, random letter insertion, name order reversal, multilingual translation perturbations, as well as OOD splits from GLUE-X. As shown in the right in Figure 5, ConCA consistently achieves superior performance across nearly all OOD settings, indicating that the representations it learns generalize more robustly under distributional shifts.

The improvements above are likely attributable to ConCA's principled framework, grounded in Theorem 2.2, which motivates theorem-driven method design by enforcing sparsity in the exponentiated feature space and leveraging normalization to avoid overfitting, thereby yielding transferable features under both few-shot and OOD scenarios.

## 5 CONCLUSION

Observing the lack of a clear theoretical understanding behind SAEs motivated us to formalize the relationship between LLM representations and human-interpretable concepts. We showed that, under mild assumptions, LLM representations can be approximated as linear mixtures of the log-posteriors of latent concepts. Building on this insight, we introduced ConCA, including a sparse variant, to recover these concept posteriors in an unsupervised manner. Empirical results across multiple models and benchmarks demonstrate that ConCA extracts features that outperform SAE variants in both faithfulness and utility. Looking forward, our framework opens the door to principled analysis, manipulation, and evaluation of LLM representations, as well as exploration of alternative regularization strategies to further enhance interpretability.

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

# Appendix

## Table of Contents

# A   RELATED WORK

**Sparse Autoencoders and Dictionary Learning**   The proposed ConCA framework is closely related to Sparse Autoencoders (SAEs) (Rajamanoharan et al., 2024a;b; Gao et al., 2025; Braun et al., 2024; Bricken et al., 2023; Huben et al., 2023; Gao et al., 2025; Mudide et al., 2024; Chanin et al., 2024; Lieberum et al., 2024; He et al., 2024; Karvonen et al., 2024; Bussmann et al., 2024), as both aim to extract and monosemantic human-interpretable concepts from LLM representations in order to provide mechanistic explanations for their success. However, the two approaches differ fundamentally in their theoretical foundations. ConCA is grounded in a rigorous theoretical framework, as established in Theorem 2.2, while SAEs rely on assumptions such as the linear representation hypothesis and the superposition hypothesis. This foundational difference leads to notable divergences in both method design and evaluation protocols, as discussed in the Introduction. In addition, our work is also closely connected to the well-established framework of dictionary learning (Dumitrescu & Irofti, 2018; Eggert & Korner, 2004; Elad, 2010; Elad & Bruckstein, 2002; Aharon et al., 2006; Arora et al., 2015). Specifically, this work bridges next-token prediction framework and dictionary learning by showing that LLM representations acquired through the next-token prediction framework can be further meaningfully decomposed using dictionary learning-like techniques.

**Causal Representation Learning**   This work is also related to causal representation learning (Schölkopf et al., 2021), which seeks to identify latent causal variables from observational data (Brehmer et al., 2022; Von Kügelgen et al., 2021; Massidda et al., 2023; von Kügelgen et al., 2023; Ahuja et al., 2023; Seigal et al., 2022; Shen et al., 2022; Liu et al., 2022; Buchholz et al., 2023; Varici et al., 2023; Liu et al., 2024c; 2025b; 2024b;a; Hyvarinen & Morioka, 2016; Hyvarinen et al., 2019; Khemakhem et al., 2020; Cai et al., 2025; Rajendran et al., 2024). Most of those works focus on continuous latent and observed variables, we explore the setting of discrete variables. A subset of studies has investigated causal representation learning in discrete spaces (Gu & Dunson, 2023; Kong et al., 2024; Kivva et al., 2021), but these typically assume specific graphical structures and rely on invertible mappings from latent to observed variables. In contrast, our approach does not require such assumptions, offering greater flexibility.

**Identifiability Analysis for LLMs**   Several prior studies (Marconato et al., 2024; Roeder et al., 2021) have explored identifiability within the inference space, revealing alignments between representations obtained from distinct inference models. However, these findings remain confined to the inference space and do not extend to identifying the true latent variables in latent variable models. More recently, Jiang et al. (2024) examined the emergence of linear structures under a different generative framework, attributing them to the implicit bias introduced by gradient-based optimization. In contrast, our approach offers a theoretical explanation rooted in identifiability theory, directly linking the observed linear patterns to the ground-truth latent structure. This shift in perspective provides a deeper and more principled understanding of the underlying mechanisms.

**Concept Discovery**   Concept discovery aims to extract human-interpretable concepts from pre-trained models, and has emerged as a key area within machine learning (Schut et al., 2023; Yang et al., 2023; Marconato et al., 2023; Oikarinen et al., 2023; Koh et al., 2020; Schwalbe, 2022; Poeta et al., 2023; Taeb et al., 2022). While empirical methods have flourished, theoretical understanding of when and how such concepts can be reliably identified remains limited. In contrast, the proposed ConCA is grounded in rigorous theoretical results. The work of Leemann et al. (2023) investigates concept identifiability under the assumption that the non-linear mapping is known a priori. In contrast, our results establish identifiability guarantees without requiring such prior knowledge. A recent advance by Rajendran et al. (2024) offers formal identifiability results for continuous latent concepts under a likelihood-matching framework, while our work focuses on discrete concepts, and approaches the problem from a different angle, i.e., rooted in the next-token prediction paradigm, which underpins modern LLMs training. This shift in both focus and framework allows us to derive new identifiability guarantees tailored to the discrete setting.

## B    LIMITATIONS AND DISCUSSION

**Theoretical limitations.**    Our theoretical analysis in Theorem 2.2 relies on several assumptions, including the Diversity Condition (i) and the Informational Sufficiency Condition (ii). While we provide justifications and argue that these assumptions are likely mild in practice, they represent idealized conditions that may not hold exactly in real-world datasets. Nonetheless, we believe they are reasonable: most have been introduced in prior work, and some, such as the Informational Sufficiency Condition (ii), are already considered relaxations in previous identifiability analyses within the causal representation learning community.

**Methodological limitations.**    In addition, sparse ConCA applies regularization to recover concept-level posteriors, but in practice we do not use the exponential function directly on $\hat{\mathbf{z}} \approx [\log p(z_i|\mathbf{x})]_{z_i}$. Instead, we employ exponential-like activation functions, such as SELU, SoftPlus, and ELU, which approximate exponential behavior for small input values while ensuring numerical stability and smooth gradients. As a result, the sparsity prior only approximately reflects true posterior activation. Additionally, underdetermined settings ($\ell > m$) and deviations from theoretical assumptions can lead to multiple plausible solutions, some capturing noise rather than meaningful concepts.

**Discussion.**    Despite the theoretical, methodological, and evaluation limitations discussed above, sparse ConCA provides a principled framework for understanding and decomposing LLM representations at the concept level. The approach highlights the potential of leveraging sparsity and structured priors to recover interpretable latent factors, even under underdetermined settings or approximate assumptions. Furthermore, our work emphasizes the importance of carefully designed evaluation frameworks, as concept-level recovery in natural language remains inherently challenging. We hope that these insights will motivate future research to incorporate additional probabilistic constraints, explore alternative regularization strategies, and develop larger and more diverse benchmarks to better evaluate and improve concept extraction methods in LLMs. Ultimately, sparse ConCA serves as a starting point for building more interpretable, reliable, and theoretically grounded tools for analyzing complex representations in LLM.

| Aspect | Our Theorem 2.2 | Liu's Theorem C.1 |
|---|---|---|
| **Diversity Condition 1** | Requires only $m + 1$ distinct $y$ values | Requires $\prod_i k_i + 1$ distinct $y$ values |
| **Diversity Condition 2** | Not required | Required |
| **Representations** | $\mathbf{A}[\ldots; [\log p(z_i|\mathbf{x})]_{z_i}; \ldots] + \mathbf{b}$ | $\mathbf{A}[\log p(\mathbf{z} = \mathbf{z}_i|\mathbf{x})]_{\mathbf{z}_i} + \mathbf{b}$ |
| **Interpretability** | Mixture of *component-wise* $p(z_j|\mathbf{x})$ | Mixture of *joint* $[\log p(\mathbf{z} = \mathbf{z}_i|\mathbf{x})]_{\mathbf{z}_i}$ |

Table 2: Comparison of Our Theorem and Liu's Theorem in terms of assumptions and results. This table highlights that our theorem requires weaker assumptions and provides more interpretable, component-wise results.

## C  COMPARISON OF THE RESULT IN LIU ET AL. (2025A)

The theoretical result Theorem 2.2 in this work is totally different with that of Liu et al. (2025a). For comparison, here we re-write the result in Liu et al. (2025a) as follows:

**Theorem C.1** (Liu et al. (2025a))**.** *Suppose latent variables $\mathbf{z}$ and the observed variables $\mathbf{x}$ and $y$ follow the generative models defined in Eq. 1, and assume that $\mathbf{z}$ takes values in a finite set of cardinality $k$. Assume the following holds:*

*(i) (**Diversity Condition 1**) There exist $\prod_i k_i + 1$ values of $y$, so that the matrix $\mathbf{L} = \big(\mathbf{g}(y = y_1) - \mathbf{g}(y = y_0), ..., \mathbf{g}(y = y_k) - \mathbf{g}(y = y_0)\big)$ of size $\prod_i k_i \times \prod_i k_i$ is invertible,*

*(ii) (**Diversity Condition 2**) There exist $k + 1$ distinct values of $y$, i.e., $y_0,...,y_k$, such that the matrix $\hat{\mathbf{L}} = \big([p(\mathbf{z} = \mathbf{z}_i|y = y_1) - p(\mathbf{z} = \mathbf{z}_i|y = y_0)]_{\mathbf{z}_i}, ..., [p(\mathbf{z} = \mathbf{z}_i|y = y_k) - p(\mathbf{z} = \mathbf{z}_i|y = y_0)]_{\mathbf{z}_i}\big)$ of the size $k \times k$ is invertible*

*(iii) (**Approximate Invertibility Condition**) The mapping from $\mathbf{z}$ to $(\mathbf{x}, y)$ is approximately invertible in the sense that the posterior $p(\mathbf{z} \mid \mathbf{x}, y)$ is sharply peaked, i.e., there exists a most probable $\mathbf{z}^*$ such that $p(\mathbf{z} = \mathbf{z}^* \mid \mathbf{x}, y) \geq 1 - \epsilon$ for some $\epsilon \in [0, 1)$ with $\epsilon \to 0$.*

*Then the true latent variables $\mathbf{z}$ are mathematically related to the representations in LLMs, i.e., $\mathbf{f}(\mathbf{x})$, which are learned through the next-token prediction framework, by the following relationship:*

$$\mathbf{f}(\mathbf{x}) \approx \mathbf{A}[\log p(\mathbf{z} = \mathbf{z}_i|\mathbf{x})]_{\mathbf{z}_i} + \mathbf{b}, \tag{6}$$

*where $\mathbf{A} = (\hat{\mathbf{L}}^T)^{-1}\mathbf{L}$, and $\mathbf{b}$ is a bias vector.*

We compare our Theorem 2.2 with Theorem C.1 mainly from the following (Also see Table 2 for a summary of the comparison.):

- Assumptions: Our Theorem 2.2 eliminates the need for Diversity Condition 2 ((ii)). Furthermore, compared to Diversity Condition 1 ((i)), our assumption requires only $m + 1$ distinct values of $y$, where $m$ is the dimensionality of the learned LLM representations. In contrast, Condition (i) requires $k + 1$ values, where $k$ is the number of possible configurations of the discrete latent variable $\mathbf{z}$ (i.e., the number of distinct values $\mathbf{z}$ can take). Since it is generally believed that $m < k$, our assumption is strictly weaker. This belief is partly supported by the commonly discussed the superposition hypothesis (Elhage et al., 2022).

- Results: Our result shows that the LLM representation $\mathbf{f}(\mathbf{x})$ approximates a mixture over the individual components of the latent variable $\mathbf{z}$, i.e., $\mathbf{f}(\mathbf{x}) \approx \mathbf{A}[\log p(z_1 \mid \mathbf{x}), \ldots, \log p(z_\ell \mid \mathbf{x})] + \mathbf{b}$, whereas Theorem C.1 describes $\mathbf{f}(\mathbf{x})$ as a mixture over the full configurations of the joint latent variable $\mathbf{z}$, i.e., $\mathbf{f}(\mathbf{x}) \approx \mathbf{A}[\log p(\mathbf{z} = \mathbf{z}_i \mid \mathbf{x})]_{\mathbf{z}_i} + \mathbf{b}$. Most importantly, our result support that one can estimate each distribution $p(z_j \mid \mathbf{x})$ by unmixing a linear combination, offering a more interpretable and component-wise understanding of the learned representation.

# D  LEMMAS IN THE CONTEXT OF $H(\mathbf{z} \mid \mathbf{x}) \to 0$

For ease of exposition in the following sections, we first introduce the following lemmas.

**Lemma D.1** (Factorization of the Posterior as Conditional Entropy Vanishes). *Suppose latent causal variables $\mathbf{z} = (z_1, \ldots, z_\ell)$ and observed variable $\mathbf{x}$ follow the causal generative model defined in Eq. 1. Then:*

$$p(\mathbf{z} \mid \mathbf{x}) \approx \prod_{i=1}^{\ell} p(z_i \mid \mathbf{x}), \quad \textit{as } H(\mathbf{z} \mid \mathbf{x}) \to 0. \tag{7}$$

**Intuition.**  When $H(\mathbf{z} \mid \mathbf{x}) = 0$, the observation $\mathbf{x}$ uniquely determines every coordinate $z_i$, so no residual dependence remains between them. If the conditional entropy is merely small, the remaining dependencies are weak and the posterior is well-approximated by $\prod_i p(z_i \mid \mathbf{x})$.

*Proof.*  Define the product of marginals as:

$$q(\mathbf{z} \mid \mathbf{x}) := \prod_{i=1}^{\ell} p(z_i \mid \mathbf{x}). \tag{8}$$

The Kullback–Leibler divergence between $p(\mathbf{z} \mid \mathbf{x})$ and $q(\mathbf{z} \mid \mathbf{x})$ is

$$D_{\mathrm{KL}}\big(p(\mathbf{z} \mid \mathbf{x}) \| q(\mathbf{z} \mid \mathbf{x})\big) = \mathbb{E}_{p(\mathbf{z}|\mathbf{x})} \left[ \log \frac{p(\mathbf{z} \mid \mathbf{x})}{\prod_{i=1}^{\ell} p(z_i \mid \mathbf{x})} \right]. \tag{9}$$

Recall that conditional entropy satisfies

$$H(\mathbf{z} \mid \mathbf{x}) = -\mathbb{E}_{p(\mathbf{z}|\mathbf{x})} \log p(\mathbf{z} \mid \mathbf{x}), \tag{10}$$

and similarly for each marginal,

$$H(z_i \mid \mathbf{x}) = -\mathbb{E}_{p(z_i|\mathbf{x})} \log p(z_i \mid \mathbf{x}). \tag{11}$$

Thus,

$$D_{\mathrm{KL}}\big(p(\mathbf{z} \mid \mathbf{x}) \| q(\mathbf{z} \mid \mathbf{x})\big) = \mathbb{E}_{p(\mathbf{z}|\mathbf{x})} \left[ \log p(\mathbf{z} \mid \mathbf{x}) - \sum_{i=1}^{\ell} \log p(z_i \mid \mathbf{x}) \right] \tag{12}$$

$$= -H(\mathbf{z} \mid \mathbf{x}) - \sum_{i=1}^{\ell} \mathbb{E}_{p(\mathbf{z}|\mathbf{x})} \left[ -\log p(z_i \mid \mathbf{x}) \right] \tag{13}$$

$$= \sum_{i=1}^{\ell} H(z_i \mid \mathbf{x}) - H(\mathbf{z} \mid \mathbf{x}), \tag{14}$$

where we have used the law of total expectation to replace $\mathbb{E}_{p(\mathbf{z}|\mathbf{x})}[-\log p(z_i \mid \mathbf{x})]$ by $H(z_i \mid \mathbf{x})$.

By the chain rule of entropy, for each $i$ we have

$$H(\mathbf{z} \mid \mathbf{x}) - H(z_i \mid \mathbf{x}) = H(\mathbf{z}_{-i} \mid z_i, \mathbf{x}), \tag{15}$$

where $\mathbf{z}_{-i}$ denotes all components except $z_i$. Since entropy is non-negative for discrete case,

$$H(\mathbf{z} \mid \mathbf{x}) \geq H(z_i \mid \mathbf{x}). \tag{16}$$

Then, if $H(\mathbf{z} \mid \mathbf{x}) \to 0$, we necessarily have

$$H(z_i \mid \mathbf{x}) \to 0, \quad \text{for all } i = 1, \ldots, \ell. \tag{17}$$

Combining the above,

$$D_{\mathrm{KL}}\big(p(\mathbf{z} \mid \mathbf{x}) \| q(\mathbf{z} \mid \mathbf{x})\big) = \sum_{i=1}^{\ell} H(z_i \mid \mathbf{x}) - H(\mathbf{z} \mid \mathbf{x}) \to 0. \tag{18}$$

$\square$

**Lemma D.2** (Exact Linear Representation of Joint Log Posterior via Full Marginals). *Let* $\mathbf{z} = (z_1, \ldots, z_\ell) \in \mathcal{V}_1 \times \cdots \times \mathcal{V}_\ell$ *to be discrete, with* $z_i \in \mathcal{V}_i, |\mathcal{V}_i| = k_i, i = 1, \ldots, \ell$, *then there exists a fixed (assignment-independent) selector matrix* $\mathbf{S} \in \{0, 1\}^{M \times \sum_i k_i}$ *with* $M = \prod_i k_i$ *such that*

$$[\log p(\mathbf{z} \mid \mathbf{x})]_{\mathbf{z}} \approx \mathbf{S} \big[ [\log p(z_1 \mid \mathbf{x})]_{z_1}; \ldots; [\log p(z_\ell \mid \mathbf{x})]_{z_\ell} \big], \tag{19}$$

*where the approximation becomes accurate as* $H(\mathbf{z} \mid \mathbf{x}) \to 0$ *(i.e., the posterior concentrates on a few high-probability assignments). The $j$-th row of* $\mathbf{S}$ *selects, for each $i$, the entry in* $\big[ [\log p(z_1 \mid \mathbf{x})]_{z_1}, , \ldots, [\log p(z_\ell \mid \mathbf{x})]_{z_\ell} \big]$ *corresponding to the value of $z_i$ in the $j$-th joint assignment* $\mathbf{z}^{(j)}$.

**Intuition.** The vector $\big[ [\log p(z_1 \mid \mathbf{x})]_{z_1}; \ldots; [\log p(z_\ell \mid \mathbf{x})]_{z_\ell} \big]$ stacks all single-variable log-posteriors. The selector matrix $\mathbf{S}$ picks, for each joint assignment, the corresponding entries in $\big[ [\log p(z_1 \mid \mathbf{x})]_{z_1}; \ldots; [\log p(z_\ell \mid \mathbf{x})]_{z_\ell} \big]$ so that $\mathbf{S} \big[ [\log p(z_1 \mid \mathbf{x})]_{z_1}; \ldots; [\log p(z_\ell \mid \mathbf{x})]_{z_\ell} \big]$ reconstructs the joint log-posterior $[\log p(\mathbf{z} \mid \mathbf{x})]_{\mathbf{z}}$. Under low conditional entropy $H(\mathbf{z} \mid \mathbf{x}) \to 0$, only a few joint assignments dominate, making this approximation accurate.

*Proof.* Let $M = \prod_i k_i$ and enumerate all joint assignments as $\mathcal{V}^\ell = \{\mathbf{z}^{(1)}, \ldots, \mathbf{z}^{(M)}\}$. Consider the vector

$$[\log p(\mathbf{z} \mid \mathbf{x})]_{\mathbf{z}} \in \mathbb{R}^M, \tag{20}$$

whose $j$-th entry is $\log p(\mathbf{z}^{(j)} \mid \mathbf{x})$.

**Step 1 (Factorization under low entropy).** By Lemma D.1, as $H(\mathbf{z} \mid \mathbf{x}) \to 0$,

$$p(\mathbf{z} \mid \mathbf{x}) \approx \prod_{i=1}^{\ell} p(z_i \mid \mathbf{x}) \quad \implies \quad \log p(\mathbf{z}^{(j)} \mid \mathbf{x}) \approx \sum_{i=1}^{\ell} \log p(z_i^{(j)} \mid \mathbf{x}) \tag{21}$$

where $z_i^{(j)}$ denotes the value of the $i$-th latent variable in the $j$-th joint assignment $\mathbf{z}^{(j)}$.

**Step 2 (Construction of selector matrix).** Construct $\mathbf{S} \in \{0, 1\}^{M \times \sum_i k_i}$ so that its $j$-th row has exactly one 1 in each block corresponding to variable $z_i$, selecting the entry that corresponds to the value of $z_i$ in $\mathbf{z}^{(j)}$, and all other entries in that row are 0.

Then for each $j$,

$$\big( \mathbf{S} \big[ [\log p(z_1 \mid \mathbf{x})]_{z_1}; \ldots; [\log p(z_\ell \mid \mathbf{x})]_{z_\ell} \big] \big)_j = \sum_{i=1}^{\ell} \log p(z_i^{(j)} \mid \mathbf{x}) \approx \log p(\mathbf{z}^{(j)} \mid \mathbf{x}), \tag{22}$$

so the linear map exactly reproduces the sum of marginal log-probabilities for the joint assignment.

**Step 3 (Validity under low entropy).** Because the posterior concentrates on a few high-probability assignments as $H(\mathbf{z} \mid \mathbf{x}) \to 0$, the sum-of-marginals approximation is accurate for the entries corresponding to these assignments. Thus

$$[\log p(\mathbf{z} \mid \mathbf{x})]_{\mathbf{z}} \approx \mathbf{S} \big[ [\log p(z_1 \mid \mathbf{x})]_{z_1}; \ldots; [\log p(z_\ell \mid \mathbf{x})]_{z_\ell} \big], \tag{23}$$

as claimed. The matrix $\mathbf{S}$ is fixed (assignment-independent) and encodes the mapping from full marginal logs to joint-log vector. $\square$

**Lemma D.3** (Expectation difference vanishes with conditional entropy). *Suppose latent causal variables* $\mathbf{z}$ *and observed variable* $\mathbf{x}$ *follow a generative model. For any two values* $y_0$ *and* $y_i$, *as* $H(\mathbf{z} \mid \mathbf{x}) \to 0$,

$$\mathbb{E}_{p(\mathbf{z}|y_i)}[\log p(\mathbf{z} \mid \mathbf{x})] - \mathbb{E}_{p(\mathbf{z}|y_0)}[\log p(\mathbf{z} \mid \mathbf{x})] \longrightarrow 0.$$

**Intuition.** When $\mathbf{x}$ almost fully determines $\mathbf{z}$, the expectation $\mathbb{E}_{p(\mathbf{z}|y)}[\log p(\mathbf{z} \mid \mathbf{x})]$ becomes nearly independent of $y$, so the difference between any two values of $y$ vanishes.

*Proof.* Let

$$\mathbf{z}^* = \arg\max_z p(z \mid \mathbf{x}), \quad \varepsilon = 1 - p(\mathbf{z}^* \mid \mathbf{x}). \tag{24}$$

Since $H(\mathbf{z} \mid \mathbf{x}) \to 0$, the conditional distribution $p(\mathbf{z} \mid \mathbf{x})$ becomes increasingly concentrated on $\mathbf{z}^*$, i.e., $\varepsilon \to 0$.

Now, for any fixed $y$, as $\varepsilon \to 0$, the posterior satisfies

$$p(\mathbf{z}^* \mid \mathbf{x}) = 1 - \varepsilon, \tag{25}$$

and for all $\mathbf{z} \neq \mathbf{z}^*$, we have

$$p(\mathbf{z} \mid \mathbf{x}) \approx \varepsilon. \tag{26}$$

We can then decompose the expectation:

$$\mathbb{E}_{p(\mathbf{z}|y)}[\log p(\mathbf{z} \mid \mathbf{x})] = p(\mathbf{z}^* \mid y) \log p(\mathbf{z}^* \mid \mathbf{x}) + \sum_{\mathbf{z} \neq \mathbf{z}^*} p(\mathbf{z} \mid y) \log p(\mathbf{z} \mid \mathbf{x}) \tag{27}$$

$$\tag{28}$$

Note that:

- $\log(1 - \epsilon) \approx 0$, when $\epsilon \to 0$,

- $\log p(\mathbf{z} \mid \mathbf{x}) \approx \log \epsilon$ for all $\mathbf{z} \neq \mathbf{z}^*$, when $\epsilon \to 0$,

- and for $\mathbf{z} \neq \mathbf{z}^*$, $p(\mathbf{z} \mid y)$ is bounded.

Hence, for any $y$, when $\epsilon \to 0$,

$$\mathbb{E}_{p(\mathbf{z}|y)}[\log p(\mathbf{z} \mid \mathbf{x}, y)] \to \log \epsilon, \tag{29}$$

which is independent of the specific value of $y$. As a result, taking the difference for two distinct values $y_0$ and $y_i$:

$$\mathbb{E}_{p(\mathbf{z}|y_i)}[\log p(\mathbf{z} \mid \mathbf{x}, y_i)] - \mathbb{E}_{p(\mathbf{z}|y_0)}[\log p(\mathbf{z} \mid \mathbf{x}, y_0)] \to 0.$$

This completes the proof. $\square$

## E    PROOF OF THEOREM 2.2

**Theorem 2.1.** *Suppose latent variables* $\mathbf{z}$ *and the observed variables* $\mathbf{x}$ *and* $y$ *follow the generative models defined in Eq.* 1. *Assume the following holds:*

(i) *(**Diversity Condition**) There exist* $m + 1$ *values of* $y$, *so that the matrix* $\mathbf{L} = \big(\mathbf{g}(y = y_1) - \mathbf{g}(y = y_0), ..., \mathbf{g}(y = y_m) - \mathbf{g}(y = y_0)\big)$ *of size* $m \times m$ *is invertible,*

(ii) *(**Informational Sufficiency Condition**) The conditional entropy of the latent concepts given the context is close to zero, i.e.,* $H(\mathbf{z}|\mathbf{x}) \to 0$,

*then the representations* $\mathbf{f}(\mathbf{x})$ *in LLMs, which are learned through the next-token prediction framework, are related to the true latent variables* $\mathbf{z}$, *by the following relationship:*

$$\mathbf{f}(\mathbf{x}) \approx \mathbf{A}\big[[\log p(z_1 \mid \mathbf{x})]_{z_1}; \ldots; [\log p(z_\ell \mid \mathbf{x})]_{z_\ell}\big] + \mathbf{b}, \tag{30}$$

*where* $\mathbf{A}$ *is a* $m \times (\sum_{i=1}^\ell k_i)$ *matrix,* $\mathbf{b}$ *is a bias vector.*

**Intuition.**    Each LLM representation $\mathbf{f}(\mathbf{x})$ encodes a combination of all latent concepts. Under the Diversity, observing representations across multiple diverse outputs $y$ provides linearly independent constraints that reveal how all latent concepts contribute together, i.e., the joint posterior. When the latent concepts are nearly determined by $\mathbf{x}$, i.e., the Informational Sufficiency Condition, the joint posterior decomposes into marginal posteriors, allowing $\mathbf{f}(\mathbf{x})$ to be expressed as a linear mixture of the log-posteriors of individual concepts.

*Proof.* Recall that next-token prediction can be viewed as a multinomial logistic regression model, where the conditional distribution is approximated as

$$p(y|\mathbf{x}) = \frac{\exp\big(\mathbf{f}(\mathbf{x})^\top \mathbf{g}(y)\big)}{\sum_{y'} \exp\big(\mathbf{f}(\mathbf{x})^\top \mathbf{g}(y')\big)}. \tag{31}$$

Here, $\mathbf{f}(\mathbf{x})$ and $\mathbf{g}(y)$ denote the learned representations of $\mathbf{x}$ and $y$, respectively, both lying in $\mathbb{R}^m$.

On the other hard, under the latent-variable formulation in Eq. 1, the conditional distribution is given by marginalization:

$$p(y|\mathbf{x}) = \sum_{\mathbf{z}} p(y|\mathbf{z}) \, p(\mathbf{z}|\mathbf{x}). \tag{32}$$

Equating Eq. 31 and Eq. 32, we obtain

$$\frac{\exp\big(\mathbf{f}(\mathbf{x})^\top \mathbf{g}(y)\big)}{\sum_{y'} \exp\big(\mathbf{f}(\mathbf{x})^\top \mathbf{g}(y')\big)} = \sum_{\mathbf{z}} p(y|\mathbf{z}) \, p(\mathbf{z}|\mathbf{x}). \tag{33}$$

Taking logarithms on both sides, one arrives at

$$\mathbf{f}(\mathbf{x})^\top \mathbf{g}(y) - \log Z(\mathbf{x}) = \log \sum_{\mathbf{z}} p(y|\mathbf{z}) \, p(\mathbf{z}|\mathbf{x}), \tag{34}$$

where $Z(\mathbf{x}) = \sum_{y'} \exp(\mathbf{f}(\mathbf{x})^\top \mathbf{g}(y'))$ is the partition function.

We now focus on the right-hand side. Using Bayes' rule and the conditional independence assumption $y \perp \mathbf{x} \mid \mathbf{z}$, we can decompose:

$$\log p(y|\mathbf{x}) = \mathbb{E}_{p(\mathbf{z}|y)}\Big[\log \frac{p(y, \mathbf{z}|\mathbf{x})}{p(\mathbf{z}|y, \mathbf{x})}\Big] \tag{35}$$

$$= \mathbb{E}_{p(\mathbf{z}|y)}[\log p(\mathbf{z}|\mathbf{x})] + \mathbb{E}_{p(\mathbf{z}|y)}[\log p(y|\mathbf{z})] - \mathbb{E}_{p(\mathbf{z}|y)}[\log p(\mathbf{z}|y, \mathbf{x})]. \tag{36}$$

Combining Eq. equation 34 and Eq. equation 36, we arrive at

$$\mathbf{f}(\mathbf{x})^\top \mathbf{g}(y) - \log Z(\mathbf{x}) = \mathbb{E}_{p(\mathbf{z}|y)}[\log p(\mathbf{z}|\mathbf{x})] - \mathbb{E}_{p(\mathbf{z}|y)}[\log p(\mathbf{z}|y, \mathbf{x})] + b_y, \tag{37}$$

where we set $b_y := \mathbb{E}_{p(\mathbf{z}|y)}[\log p(y|\mathbf{z})]$ for notational convenience.

For concreteness, let $y_0, y_1, \ldots, y_m$ denote the outcomes satisfying the diversity condition in condition (i). In particular, for $y = y_0$ we have

$$\mathbf{f}(\mathbf{x})^\top \mathbf{g}(y_0) - \log Z(\mathbf{x}) = \sum_{\mathbf{z}} p(\mathbf{z}|y_0) \log p(\mathbf{z}|\mathbf{x}) - h_{y_0} + b_{y_0}, \tag{38}$$

with $h_{y_0} := \sum_{\mathbf{z}} p(\mathbf{z}|y_0) \log p(\mathbf{z}|y_0, \mathbf{x})$. Similarly, for $y = y_1$,

$$\mathbf{f}(\mathbf{x})^\top \mathbf{g}(y_1) - \log Z(\mathbf{x}) = \sum_{\mathbf{z}} p(\mathbf{z}|y_1) \log p(\mathbf{z}|\mathbf{x}) - h_{y_1} + b_{y_1}. \tag{39}$$

Subtracting Eq. equation 38 from Eq. equation 39, we obtain

$$\big(\mathbf{g}(y_1) - \mathbf{g}(y_0)\big)^\top \mathbf{f}(\mathbf{x}) = \Big( \sum_{\mathbf{z}} \big(p(\mathbf{z}|y_1) - p(\mathbf{z}|y_0)\big) \log p(\mathbf{z}|\mathbf{x}) \Big) - (h_{y_1} - h_{y_0}) + (b_{y_1} - b_{y_0}). \tag{40}$$

Since $y$ can take $m + 1$ distinct values, Eq. 40 yields $m$ linearly independent equations. Collecting them together, we obtain

$$\underbrace{\big(\mathbf{f}_y(y_1) - \mathbf{f}_y(y_0), \ldots, \mathbf{f}_y(y_\ell) - \mathbf{f}_y(y_0)\big)^\top}_{\mathbf{L}^\top} \mathbf{f}_\mathbf{x}(\mathbf{x}) \tag{41}$$

$$= \underbrace{\big([p(\mathbf{z}|y_1) - p(\mathbf{z}|y_0)]_\mathbf{z}, \ldots, [p(\mathbf{z}|y_\ell) - p(\mathbf{z}|y_0)]_\mathbf{z}\big)^\top}_{\hat{\mathbf{L}}} [\log p(\mathbf{z}|\mathbf{x})]_\mathbf{z}$$

$$- \underbrace{[h_{y_1} - h_{y_0}, \ldots, h_{y_\ell} - h_{y_0}]}_{\mathbf{h}_y} + \underbrace{[b_{y_1} - b_{y_0}, \ldots, b_{y_\ell} - b_{y_0}]}_{\mathbf{b}_y}. \tag{42}$$

By the diversity condition, the matrix $\mathbf{L} \in \mathbb{R}^{m \times m}$ is invertible. Hence we can solve for $\mathbf{f}_\mathbf{x}(\mathbf{x})$:

$$\mathbf{f}_\mathbf{x}(\mathbf{x}) = (\mathbf{L}^\top)^{-1} \hat{\mathbf{L}} [\log p(\mathbf{z}|\mathbf{x})]_\mathbf{z} - (\mathbf{L}^\top)^{-1} \mathbf{h}_y + \underbrace{(\mathbf{L}^\top)^{-1} \mathbf{b}_y}_{\mathbf{b}}. \tag{43}$$

Then, as $H(\mathbf{z} \mid \mathbf{x}) \to 0$, by lemmas D.2 and D.3, we have:

$$\mathbf{f}_\mathbf{x}(\mathbf{x}) = (\mathbf{L}^\top)^{-1} \hat{\mathbf{L}} \underbrace{\mathbf{S} \big[[\log p(z_1 \mid \mathbf{x})]_{z_1}; \ldots; [\log p(z_\ell \mid \mathbf{x})]_{z_\ell}\big]}_{\text{by Lemma C.2}} - \underbrace{(\mathbf{L}^\top)^{-1} \mathbf{h}_y}_{\text{by Lemma C.3}, \to 0} + \mathbf{b}. \tag{44}$$

$\square$

Finally, defining $\mathbf{A} = (\mathbf{L}^\top)^{-1} \hat{\mathbf{L}} \mathbf{S}$ completes the proof.

# F    JUSTIFICATION FOR LOG-POSTERIOR ESTIMATION VIA LINEAR PROBING

**Corollary 3.1.** *Suppose Theorem 2.1 holds, i.e.,*

$$\mathbf{f}(\mathbf{x}) \approx \mathbf{A} \big[ \left[ \log p(z_1 \mid \mathbf{x}) \right]_{z_1}; \ldots ; \left[ \log p(z_\ell \mid \mathbf{x}) \right]_{z_\ell} \big] + \mathbf{b}. \tag{45}$$

*Let $\mathbf{x}_0$ and $\mathbf{x}_1$ be two counterfactual samples that differ only in the $i$-th latent concept $z_i$, each with its own ground-truth label. Then the corresponding representations $(\mathbf{f}(\mathbf{x}_0), \mathbf{f}(\mathbf{x}_1))$ are linearly separable with respect to these labels. In particular, there exists a weight matrix $\mathbf{W}$ such that $\mathbf{W} \tilde{\mathbf{A}}^{(i)} \approx \mathbf{I}$, and the associated logits recover the marginal posterior $[p(z_i \mid \mathbf{x})]_{z_i}$ over all possible values of $z_i$. In the context where $z_i$ is binary, the logits reduce to a two-dimensional vector, and the softmax recovers the marginal posterior $p(z_i = 0 \mid \mathbf{x})$, or equivalently, $p(z_i = 1 \mid \mathbf{x}) = 1 - p(z_i = 0 \mid \mathbf{x})$.*

**Intuition.**    The key idea is that each latent concept contributes to the representation along a distinct linear direction. Changing only one concept shifts the representation along its direction, so a simple linear classifier can isolate this change and recover the marginal posterior. For binary concepts, this reduces to a one-dimensional separation, while for multi-class concepts, each class corresponds to its own direction.

*Proof.*    Consider the approximation

$$\mathbf{f}(\mathbf{x}) \approx \mathbf{A}\mathbf{g}(\mathbf{x}) + \mathbf{b}, \quad \mathbf{g}(\mathbf{x}) = \big[ \left[ \log p(z_1 \mid \mathbf{x}) \right]_{z_1}; \ldots ; \left[ \log p(z_\ell \mid \mathbf{x}) \right]_{z_\ell} \big].$$

For the counterfactual samples $\mathbf{x}_0$ and $\mathbf{x}_1$ differing only in $z_i$, we pass the representations into a linear classifier with weights $\mathbf{W}$. The classifier produces logits

$$\mathbf{logits} \approx \mathbf{W}\big(\mathbf{A}\mathbf{g}(\mathbf{x}) + \mathbf{b}\big), \tag{46}$$

where **logits** is a vector over all possible values of $z_i$. In the binary case, this is a two-dimensional vector

$$[\log p(z_i = 0 \mid \mathbf{x}), \log p(z_i = 1 \mid \mathbf{x})]^\top,$$

and in the multi-class case, it contains one entry per category.

For correct classification under cross-entropy loss, the logits should recover the log-posterior for all categories (up to an additive constant):

$$\mathbf{logits} = \big[ \log p(z_i = k \mid \mathbf{x}) \big]_k + \text{const}, \tag{47}$$

where $k$ indexes all possible values of $z_i$, and the constant does not affect the softmax output.

Comparing equation 46 and equation 47, we require

$$\mathbf{W} \tilde{\mathbf{A}}^{(i)} \approx \mathbf{I}, \tag{48}$$

where $\tilde{\mathbf{A}}^{(i)}$ is the block of columns of $\mathbf{A}$ associated with all possible values of $z_i$. This condition ensures that the classifier isolates the contribution from $z_i$ and produces the correct logits.

**Binary case:** When $z_i$ is binary, $\tilde{\mathbf{A}}^{(i)}$ has two columns, and the logits reduce to a two-dimensional vector $[\log p(z_i = 0 \mid \mathbf{x}), \log p(z_i = 1 \mid \mathbf{x})]^\top$. After softmax, we directly obtain the marginal posterior

$$p(z_i = 0 \mid \mathbf{x}), \qquad p(z_i = 1 \mid \mathbf{x}) = 1 - p(z_i = 0 \mid \mathbf{x}).$$

Therefore, the counterfactual pair $(\mathbf{x}_0, \mathbf{x}_1)$ is linearly separable with respect to their ground-truth labels in both the general multi-class and binary cases.    □

# G EXPERIMENTAL DETAILS

**Training Data**  For all experiments, we use pre-trained LLMs downloaded from https://huggingface.co/, including Pythia-70m, 1.4b, 2.8b (Biderman et al., 2023), Gemma3-1b (Team et al., 2025), and Qwen3-1.7b (Team, 2025). LLM representations are extracted from these models using the first 200 million tokens of the Pile dataset, obtained from https://huggingface.co/datasets/EleutherAI/the_pile_deduplicated (Gao et al., 2020). For each token, we record the corresponding representation from the model's last hidden layer, aligned with Theorem 2.2. These pre-extracted representations form the training data for the proposed sparse ConCA and SAEs.

Table 3: Counterfactual concept pairs used for evaluation, adapted from Park et al. (2023).

| # | Concept | Example | Word Pair Counts |
|---|---------|---------|------------------|
| *Verb inflections* | | | |
| 1 | verb $\longrightarrow$ 3pSg | (accept, accepts) | 50 |
| 2 | verb $\longrightarrow$ Ving | (add, adding) | 50 |
| 3 | verb $\longrightarrow$ Ved | (accept, accepted) | 50 |
| 4 | Ving $\longrightarrow$ 3pSg | (adding, adds) | 50 |
| 5 | Ving $\longrightarrow$ Ved | (adding, added) | 50 |
| 6 | 3pSg $\longrightarrow$ Ved | (adds, added) | 50 |
| 7 | verb $\longrightarrow$ V + able | (accept, acceptable) | 50 |
| 8 | verb $\longrightarrow$ V + er | (begin, beginner) | 50 |
| 9 | verb $\longrightarrow$ V + tion | (compile, compilation) | 50 |
| 10 | verb $\longrightarrow$ V + ment | (agree, agreement) | 50 |
| *Adjective transformations* | | | |
| 11 | adj $\longrightarrow$ un + adj | (able, unable) | 50 |
| 12 | adj $\longrightarrow$ adj + ly | (according, accordingly) | 50 |
| 21 | adj $\longrightarrow$ comparative | (bad, worse) | 87 |
| 22 | adj $\longrightarrow$ superlative | (bad, worst) | 87 |
| 23 | frequent $\longrightarrow$ infrequent | (bad, terrible) | 86 |
| *Size, thing, noun* | | | |
| 13 | small $\longrightarrow$ big | (brief, long) | 25 |
| 14 | thing $\longrightarrow$ color | (ant, black) | 50 |
| 15 | thing $\longrightarrow$ part | (bus, seats) | 50 |
| 16 | country $\longrightarrow$ capital | (Austria, Vienna) | 158 |
| 17 | pronoun $\longrightarrow$ possessive | (he, his) | 4 |
| 18 | male $\longrightarrow$ female | (actor, actress) | 52 |
| 19 | lower $\longrightarrow$ upper | (always, Always) | 73 |
| 20 | noun $\longrightarrow$ plural | (album, albums) | 100 |
| *Language translations* | | | |
| 24 | English $\longrightarrow$ French | (April, avril) | 116 |
| 25 | French $\longrightarrow$ German | (ami, Freund) | 128 |
| 26 | French $\longrightarrow$ Spanish | (annee, año) | 180 |
| 27 | German $\longrightarrow$ Spanish | (Arbeit, trabajo) | 228 |

**Testing Data for Results in Figures 2 and 3.**  For evaluation, we use counterfactual text pairs that differ in only a single concept while keeping all other aspects unchanged. We emphases again that constructing such pairs is challenging due to the complexity of natural language, as also highlighted in Park et al. (2023); Jiang et al. (2024). We adopt 27 high-precision counterfactual concepts from Park et al. (2023), derived from the Big Analogy Test dataset (Gladkova et al., 2016), as our testing dataset. Table 3 lists the 27 concepts, one illustrative pair per concept, and the number of pairs used for evaluation. Despite its modest size, this benchmark suffices to meaningfully distinguish method performance and validate the sensitivity of our evaluation framework.

---

**Algorithm 1** Evaluation of SAE/ConCA Concepts via Supervised Linear Classification

---

**Require:** Trained SAEs/ConCA, 27 counterfactual pairs $\{\mathbf{x}_i\}_{i=1}^{27}$
**Ensure:** Mean Pearson correlation between SAE features and concept logits
1: **Step 1: Obtain concept logits**
2: **for** $i = 1$ **to** 27 **do**
3:     Train LogisticRegression on the $i$-th counterfactual pair
4:     Compute logit $s_i = \text{logit}(c^k = 1 | \mathbf{f}(\mathbf{x}_i))$
5: **end for**
6: Stack logits to form $\mathbf{s} = (s_1, s_2, \ldots, s_{27})$
7: **Step 2: Extract SAEs and ConCA latent features**
8: **for** $i = 1$ **to** 27 **do**
9:     Pass $\mathbf{f}(\mathbf{x}_i)$ through SAE to get latent $\hat{\mathbf{z}}_i$
10:    Compute element-wise exponentiation $\tilde{\mathbf{z}}_i = \exp(\hat{\mathbf{z}}_i)$
11: **end for**
12: Stack features to form $\tilde{\mathbf{z}} \in \mathbb{R}^{27 \times D}$
13: **Step 3: Compute correlation matrix**
14: **for** $d = 1$ **to** $D$ **do**
15:    Compute Pearson correlation $R_d = \text{C}(\mathbf{s}, \tilde{\mathbf{z}}_{:,d})$ ($D$ denotes SAEs/ConCA's feature dimension)
16: **end for**
17: **Step 4: Solve assignment problem**
18: Apply Hungarian algorithm on $\mathbf{R}$ to obtain optimal assignment.
19: Compute assigned Pearson correlations
20: **Step 5: Aggregate metric**
21: Report mean Pearson correlation across the 27 concepts.

---

**Testing Data for Downsteam Tasks in Figure 5.** For the few-show learning setting in Figure 5, we leverage a previously collected set of 113 binary classification datasets from Kantamneni et al. (2025), covering diverse tasks including challenging cases such as front-page headline detection and logical entailment. Each dataset provides prompts and binary targets (0 or 1), with prompt lengths ranging from 5 to 1024 tokens. Refer to Table 3 in Kantamneni et al. (2025) for details. For the out-of-distribution task in Figure 5, we also leverage 8 datasets from Kantamneni et al. (2025). These include: These include: 2 preexisting GLUE-X datasets designed as "extreme" versions of tasks testing grammaticality and logical entailment, 3 datasets with altered language, i.e., Tanslated to Frech, Spanish, and German, and 3 datasets with syntactic modifications substitutions of names (Fictional Characters, Random Letter Inserted, and Reversed Name Order) with cartoon characters. Probes are trained in standard settings and evaluated on these out-of-distribution test examples. Both two dataset can be downloaded from https://github.com/JoshEngels/SAE-Probes/tree/main.

**Training Pipeline.** All ConCA and SAE variants use a feature dimension of $2^{15}$, based on empirical settings from sparse SAEs. They are trained for 20,000 optimization steps with a batch size of 10,000, using the Adam optimizer with an initial learning rate of $1 \times 10^{-4}$ and a linear warm-up over the first 200 steps. For the top-$k$ and batch-top-$k$ SAEs, $k$ is set to 32. P-annealing SAEs incorporate a sparsity warm-up of 400 steps with an initial sparsity penalty coefficient 0.1. All experiments are run on a server equipped with 4 NVIDIA A100 GPUs.

**Pearson correlation coefficient** We use the PCC as the evaluation metric, as described in experiments. Algorithm 1 summarizes the procedure: for each of the 27 counterfactual concept pairs, we first obtain concept logits using a supervised linear classifier. The same inputs are then passed through the trained SAE to extract latent features, which are exponentiated and stacked into a feature matrix. We compute the Pearson correlation between each SAE feature and the corresponding concept logit, and solve the assignment problem using the Hungarian algorithm to account for permutation indeterminacy. The mean Pearson correlation across all concepts is reported as the final evaluation score.

**AUC.** The Area Under the Curve (AUC) is used to evaluate the performance of a binary classifier for concept prediction. The ROC curve plots the True Positive Rate (TPR) against the False Positive

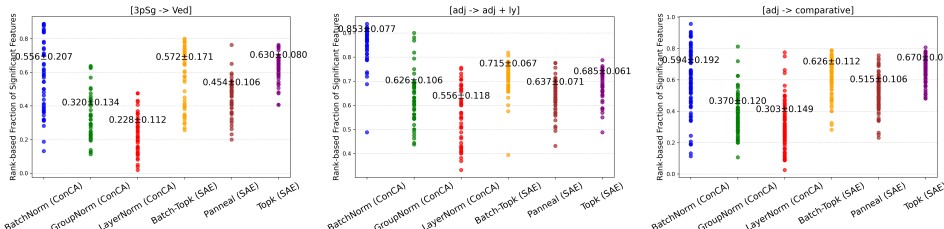

Figure 6: Rank-based fraction of significant features for SAE and ConCA variants across three counter-factual pair concepts: [3pSg – Ved], [adj – adj + ly], and [adj – comparative]. For each concept, the top-32 features with highest differences are selected, and points show the fraction of features exhibiting significant changes (mean ± std across seeds). Higher fractions indicate more variant features.

Rate (FPR) at different threshold levels, illustrating the trade-off between correctly predicting positive instances and incorrectly predicting negative instances. The AUC measures the total area under this curve, ranging from 0 to 1. An AUC of 1.0 indicates perfect classification, 0.5 corresponds to random guessing, and values closer to 0 indicate poor performance. This metric provides an aggregate, threshold-independent measure of the classifier's ability to discriminate between the two classes.

**Linear Probing in Downstream Tasks.** In our few-shot experiments, we first apply different SAEs and ConCA variants to the representations of pretrained LLMs to obtain feature embeddings. We then train a logistic regression classifier (using the LogisticRegression implementation from the scikit-learn package) on these features with limited labeled examples. To mitigate overfitting given the high dimensionality of the features, we employ an L2 penalty and select the regularization strength through cross-validation. In our out-of-distribution shift experiments, we again train a logistic regression classifier on the extracted features. To avoid overfitting to the in-distribution validation split, we fix the regularization strength to its default value (i.e., C=1.0 in LogisticRegression) instead of tuning it via cross-validation. This ensures a fairer and more stable evaluation under distribution shift.

## H    VISUALIZATION OF COUNTERFACTUAL PAIR EXPERIMENTS

To more clearly illustrate the advantages of the proposed ConCAs, we perform a visualization analysis based on features extracted by ConCA and SAE variants from counterfactual pairs. Specifically, we first select the top 32 ($k$ in top-$k$) features with the highest average absolute difference between a counterfactual pair, focusing on the most significant variations while avoiding dilution from less responsive features. We then compute the rank-based fraction of significant features over these 32 features across multiple thresholds to ensure robustness. *The fraction measures the proportion of selected features exhibiting significant changes, providing a metric of feature sensitivity and stability in response to a single concept change. Intuitively, if the learned features are expected to capture a single concept as much as possible, their responses should be small—that is, the fraction will be low.* The results are visualized using scatter plots with mean and standard deviation to capture both distribution and central tendency. This analysis highlights how ConCAs capture selective and meaningful feature variations, complementing the quantitative metrics reported earlier. For GroupNorm, we apply the same procedure independently within each group. That is, the 32 selection and rank-based fraction calculation are performed per group, and the final metric is obtained by averaging across all groups. This ensures that group-level normalization does not interfere across groups, making the evaluation consistent for GroupNorm settings. Algorithm 2 summarizes the procedure.

Figures 4, 6–13 show the rank-based fraction of significant features for the proposed ConCA and SAE variants across 27 counterfactual pairs, on Pythia-2.8b. The LayerNorm configuration exhibits the smallest fractions, indicating more stable or less variant feature changes under counterfactual conditions. In contrast, the Batch-Topk and Topk SAE variants produce larger fractions, reflecting more variable feature responses. BatchNorm and Panneal configurations display similar intermediate behavior. Overall, these trends are broadly aglined with the MPC results shown in Figure 3.

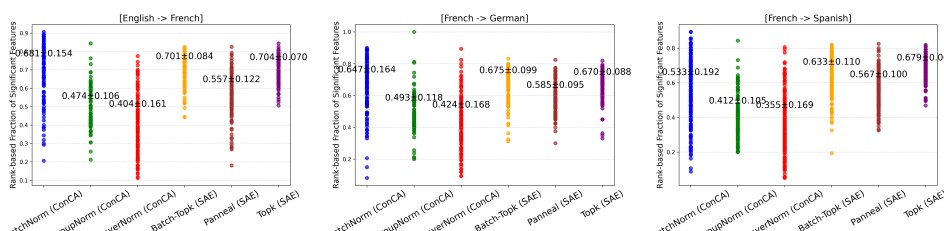

Figure 7: Rank-based fraction of significant features for SAE and ConCA variants across three counterfactual pair concepts: [English – French], [French – German], and [French – Spanish].

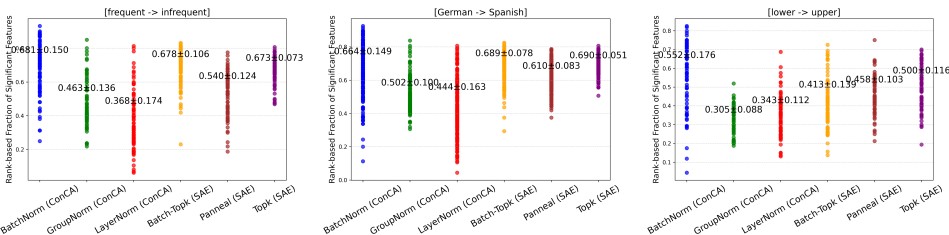

Figure 8: Rank-based fraction of significant features for SAE and ConCA variants across three counterfactual pair concepts: [frequent – infrequent], [German – Spanish], and [lower – upper].

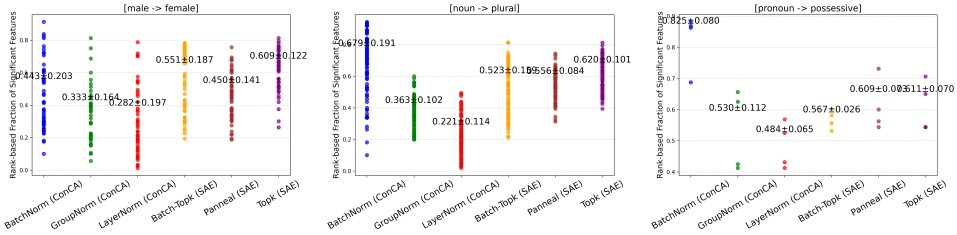

Figure 9: Rank-based fraction of significant features for SAE and ConCA variants across three counterfactual pair concepts: [male – female], [noun – plural], and [pronoun – possessive].

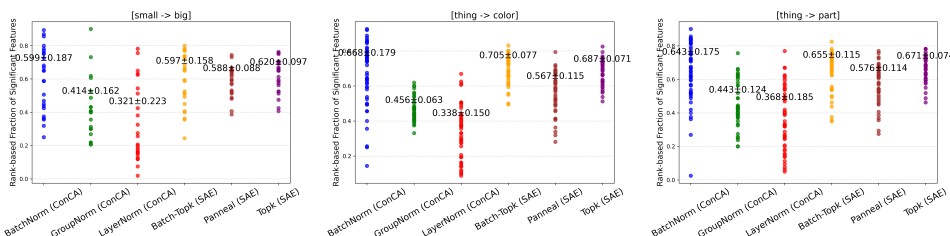

Figure 10: Rank-based fraction of significant features for SAE and ConCA variants across three counterfactual pair concepts: [small – big], [thing – color], and [thing – part].

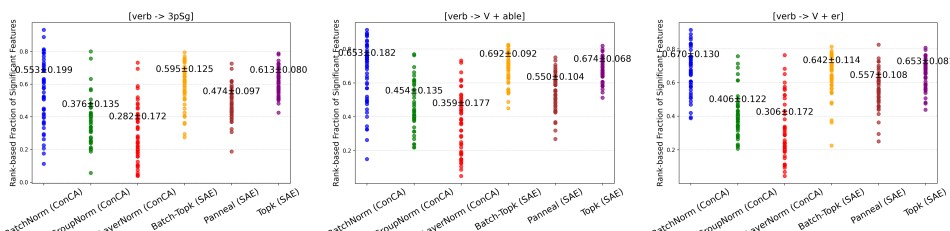

Figure 11: Rank-based fraction of significant features for SAE and ConCA variants across three counterfactual pair concepts: [verb − 3pSg], [verb − V + able], and [verb − V + er].

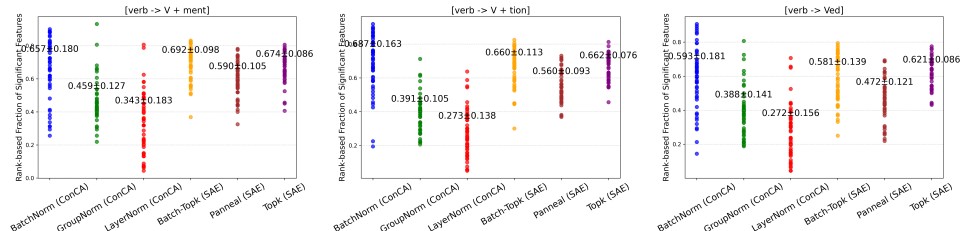

Figure 12: Rank-based fraction of significant features for SAE and ConCA variants across three counterfactual pair concepts: [verb − V + ment], [verb − V + tion], and [verb − Ved].

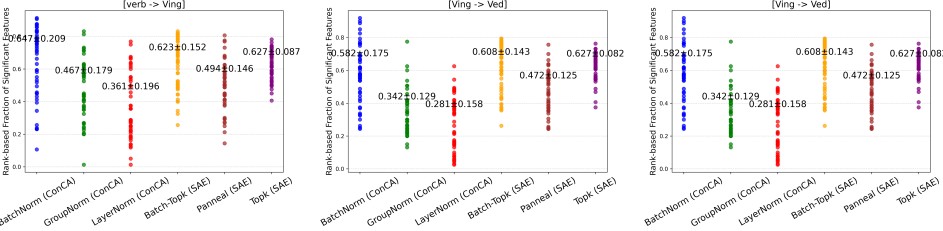

Figure 13: Rank-based fraction of significant features for SAE and ConCA variants across three counterfactual pair concepts: [verb − Ving], [Ving − Ved], and [Ving − Ved].

---

**Algorithm 2** Compute Rank-Based Fraction of Significant Features for Counterfactual Pairs

---

**Require:** Features of counterfactual pair $(\mathbf{z}_s, \mathbf{z}_t)$, $k$, `normalization`, number of groups $G$
**Ensure:** Fraction of significant features
 1: Compute element-wise absolute difference: $\mathbf{diff} = |\mathbf{z}_s - \mathbf{z}_t|$
 2: **if** `normalization` **==** "Group" **then**
 3:     Split $\mathbf{diff}, \mathbf{z}_s, \mathbf{z}_t$ into $G$ groups
 4:     **for** $g = 1$ **to** $G$ **do**
 5:         Select top-$k_g = \max(1, k/G)$ elements in group $g$ based on $\mathbf{diff}$
 6:         Convert group features $\mathbf{z}_s^g, \mathbf{z}_t^g$ to percentile ranks
 7:         Compute absolute rank differences for top-$k_g$ elements
 8:         Evaluate significance across multiple thresholds $T = \{0.1, 0.2, 0.3, 0.4, 0.5\}$
 9:         Compute average fraction of significant features for group $g$
10:     **end for**
11:     Average fractions across groups to obtain overall fraction
12: **else**
13:     Select top-$k$ elements globally based on $\mathbf{diff}$
14:     Convert selected features $\mathbf{z}_s, \mathbf{z}_t$ to percentile ranks
15:     Compute absolute rank differences for top-$k$ elements
16:     Evaluate significance across multiple thresholds $T = \{0.1, 0.2, 0.3, 0.4, 0.5\}$
17:     Compute average fraction of significant features
18: **end if**
19: **return** Overall fraction of significant features

---

## I    VISUALIZATION OF CLASSIFICATION TASKS

To better understand why ConCA features transfer effectively under the few-shot setting, we visualize the features extracted by the proposed ConCA and SAEs variants from test data using t-SNE. By projecting all features into 2D space, we can observe the structure and separability of learned representations, providing intuition for the superior downstream performance of ConCA compared to SAE variants. Figure 14 provides t-SNE visualization (Maaten & Hinton, 2008) of features of testing data, extracted by SAE and ConCA variants, on a example of the few-shot task, which shows that ConCA configurations (e.g., LayerNorm, BatchNorm, GroupNorm) produce more compact and well-separated clusters, indicating more stable and discriminative representations compared to SAE variants.

## J    ACKNOWLEDGMENT OF LLMS

We acknowledge that large language models (LLMs) were used in this work only for word-level tasks, including correcting typos, improving grammar, and refining phrasing. No substantive content, results, or scientific interpretations were generated by LLMs. All scientific ideas, analyses, and conclusions presented in this manuscript are solely the work of the authors.

## K    PRACTICAL DIAGNOSTIC FOR THE DIVERSITY CONDITION

The diversity condition in our theory requires that the model's output space contains enough linearly independent directions. Although the assumption is existential (it only requires that there exists such a set of output tokens), it is useful to provide an empirical procedure showing that modern LLMs indeed offer sufficiently diverse outputs. Since exhaustively checking all possible token combinations is infeasible, we design a practical proxy that searches for a diverse subset of outputs.

We begin by randomly sampling a large set of candidate tokens from the model's vocabulary. One token is chosen as a reference output. For each remaining candidate, we compute the difference between its output embedding and that of the reference token. These differences represent all available output directions relative to the reference. To find a subset of tokens whose directions are as independent as possible, we apply a greedy selection procedure based on LU decomposition with pivoting. This method reorders the candidate directions in decreasing order of their contribution to

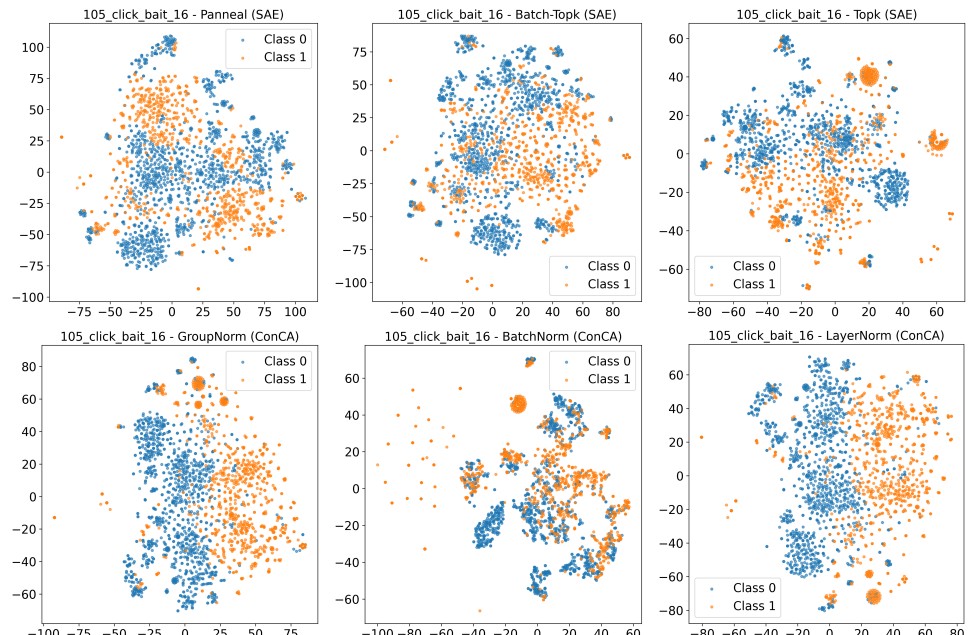

Figure 14: Visualization of features extracted by SAE and ConCA variants on a example of few-shot classification task datasets. Each point represents a test sample, colored by its class label. ConCA configurations (e.g., LayerNorm, BatchNorm, GroupNorm) produce more compact and well-separated clusters, indicating more stable and discriminative representations compared to SAE variants.

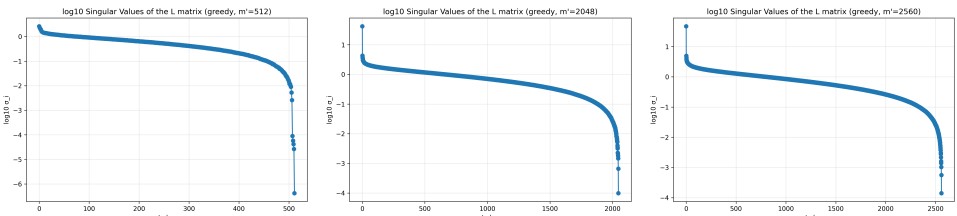

Figure 15: Singular-value spectra of the greedily selected output-difference matrix across Pythia-70m, Pythia-1.4B, and Pythia-2.8B. Each curve plots the log-scale singular values (largest to smallest) obtained from a large candidate pool of output tokens using LU-pivoting selection. The spectra decay smoothly and only collapse in the final few dimensions, indicating that each model provides a numerically full-rank set of output directions and that the diversity assumption can be approximately satisfied in practice.

the overall dimensionality. Taking the first 2,048 directions (for Pythia-12B) yields a set of outputs that spans the most independent subspace available within the candidate pool. We then assess how diverse this selected set actually is by measuring the spectrum of singular values derived from the chosen directions. A well-spread singular-value spectrum indicates that the selected outputs span a nearly full-dimensional space.

Across all model sizes, the spectra exhibit a consistent pattern. The largest several hundred singular values remain high and decay smoothly, indicating that each model provides a substantial number of independent output directions. As the index approaches the effective dimensionality of the model, the tail of the spectrum gradually drops, but only the final few singular values approach very small magnitudes. This behavior suggests that the selected output directions nearly span the model's representational space and are far from the rank-deficient structure we observe when tokens are selected at random.

Interestingly, the point at which the spectrum begins to decline sharply shifts with model scale (See Figure 15): larger models (1.4B and above) maintain strong singular values for a greater proportion

of directions compared to the 70m variant. This reflects the natural trend that larger models encode richer and more varied output embeddings. Nevertheless, even the 70m model remains numerically full-rank down to its final few dimensions.

## L    Informational Sufficiency Diagnostics

Our theory requires an informational sufficiency condition: the learned representation should contain enough information to reliably determine the related latent concept. For binary latent factors, this means the posterior distribution over the factor should be sharply peaked given the representation. Although this assumption is mild and standard in representation analyses, it is useful to provide a practical diagnostic showing that LLMs indeed satisfy it. We use the constructed 27 counterfactual concept pairs as mentioned in Table 3. These pairs cover a broad variety of concept types while keeping each concept operationally well-defined.

For each concept pair, we gather a list of word pairs exhibiting the target transformation. We extract last-token representations from the model for each word, forming two embedding sets corresponding to the two concept values. We then train a linear probe (logistic regression) to classify the target concept using 70/30 train–test splits repeated with three random seeds. The linear probe is intentionally simple: rather than maximising accuracy, its role is to estimate an empirical distribution $p(\mathbf{z}|\mathbf{x})$, allowing us to measure the uncertainty the representation leaves about the concept.

To quantify how well the representation specifies the latent concept, we compute the conditional entropy of the probe's predicted distribution over the concept: entropy near 0 bits indicates that the representation almost perfectly determines the concept; entropy near 1 bit corresponds to complete uncertainty (uniform prediction).

Across the 27 evaluated concepts (See Figure 16), we observe a clear and consistent trend: larger Pythia models yield substantially lower conditional entropy, indicating increasingly informative representations. The 70M model shows moderate informational sufficiency (typically 0.1–0.3 bits, with a few harder concepts higher), while the 1.4B and 2.8B models exhibit uniformly low entropies (mostly below 0.15 bits). This pattern suggests that, at the model scales relevant for our theoretical analysis, the representations almost deterministically encode the target concept. Consequently, the informational-sufficiency (approximate invertibility) assumption is empirically well supported.

## M    Synthetic validation of marginal vs. joint identifiability.

To complement our theoretical discussion in Appendix C, and also compare our ConCA and SAEs, we design a synthetic experiment as follow. We begin by sampling 5 binary latent variables whose causal dependencies follow an Erdős–Rényi (ER) random directed acyclic graph (DAG). Each graph is drawn with an expected number of edges equal to 10. For every node in the DAG, we define a conditional distribution over its parents using a Bernoulli model whose parameters are sampled uniformly from the interval [0.2,0.8]. We also generate counterfactual data that only in one latent variable while keeping the remains unchanged, to train base model and also train linear probing as evaluation. To simulate a nonlinear mixture process, we then convert the latent variable samples into one-hot format and randomly apply a permutation matrix to the one-hot encoding, generating one-hot observed samples. These are then transformed into binary observed samples. To simulate next-token prediction, we randomly mask a part of the binary observed data, e.g.,, $x_i$, and predict it by use the remaining portion $\mathbf{x}_{\backslash i}$. After training, we obtain the learned representations. Note that we set the representation dimension to 10 (i.e., component-wise posterior corresponds to $2 \times 5$) in order to highlight the difference between our theoretical result and the formulation in (Liu et al., 2025a), which requires a representation dimension of 32 (i.e., joint posterior corresponds to $2^5$).

Given the above setup, our first experiment aims to highlight the difference between our theoretical result and the formulation in (Liu et al., 2025a). To this end, we train a linear probe on the learned representations (as justified by Corollary 3.1). The probe achieves a classification accuracy of 0.923 (std: 0.021). This demonstrates that $2 \times 5$ dimensional representations are already sufficient for linear classification, and that $2^5$ dimensional representations required by Liu et al. (2025a) are unnecessary, thereby empirically supporting our theoretical result in Theorem 2.2.

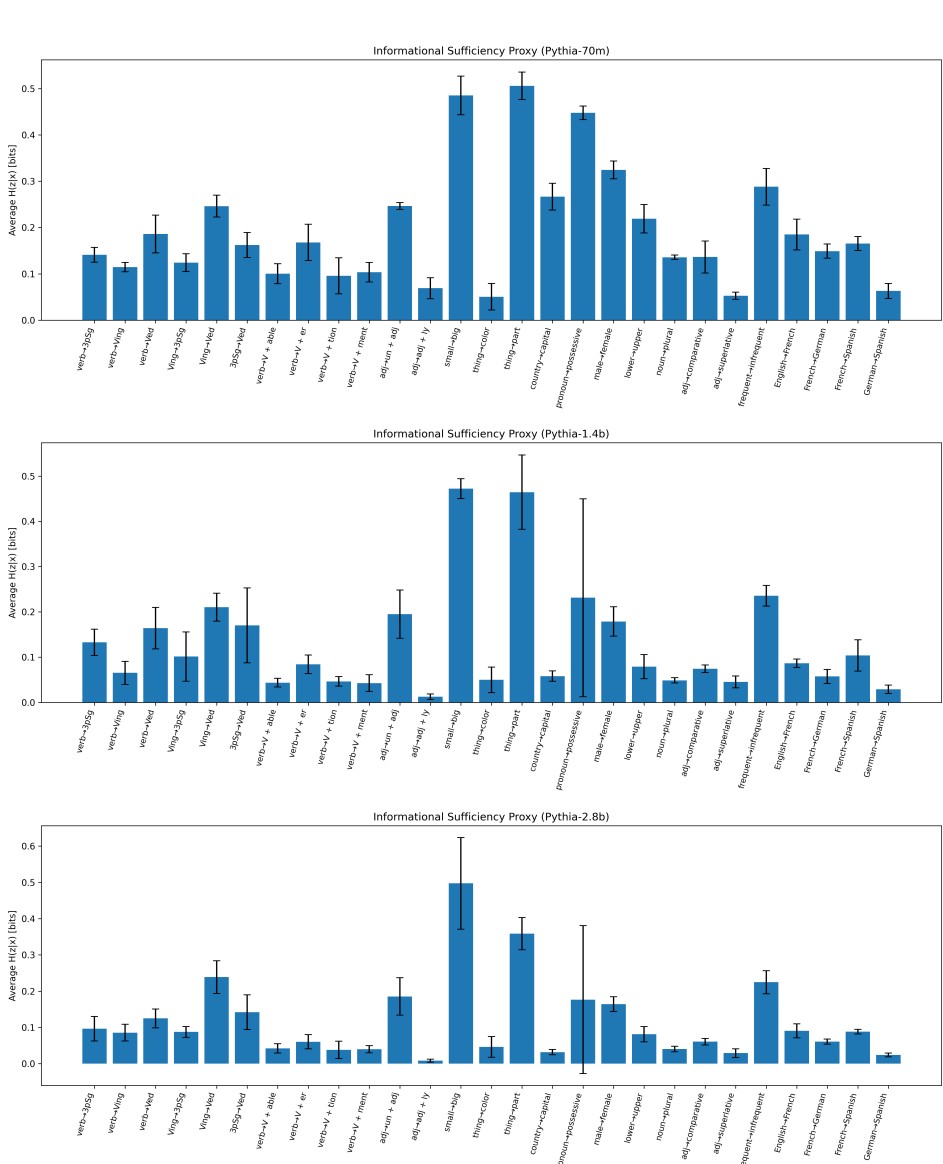

Figure 16: Conditional entropy for the 27 concept pairs across three Pythia model scales (70m, 1.4b, 2.8b). Lower entropy indicates that the representation more sharply determines the underlying concept. As model size increases, entropies consistently decrease—especially for non-trivial semantic contrasts—illustrating that informational sufficiency (approximate invertibility) emerges naturally.

| Activation Type | Clamp Range | Pearson ↑ | MSE ↓ |
|---|---|---|---|
| Exp | $[-20, 20]$ | $0.7029 \pm 0.0081$ | $10.59 \pm 2.10$ |
| Exp | $[-30, 30]$ | $0.7055 \pm 0.0148$ | $11.18 \pm 0.85$ |
| Exp | $[-40, 40]$ | $0.7020 \pm 0.0077$ | $11.03 \pm 0.97$ |
| Exp | $[-50, 50]$ | $0.7006 \pm 0.0018$ | $10.65 \pm 0.55$ |
| **SoftPlus (ours)** | — | $0.7324 \pm 0.0002$ | $7.63 \pm 0.40$ |

Table 4: Performance comparison of exponential and SoftPlus activation variants under clamped toy settings. SoftPlus achieves consistently higher Pearson correlation and lower MSE, even when the exponential is artificially stabilized.

Our second experiment is to verify the performance of our ConCA and SAEs on the learned representations. To this end, we train SAEs (Panneal), and ConCA (LayerNorm). We compute Pearson correlation between logits obtained by linear probing trained on counterfactual pair, and features obtained by SAEs and ConCA. The results are as follows: ConCA: 0.813 (std: 0.031) SAEs: 0.635 (std: 0.046). We emphasize that the scope of this work focuses on the theoretical guarantee in Theorem 2.1, which addresses identifiability up to the linear mixtures. Achieving unique recovery of individual concepts is substantially challenging. While additional assumptions, such as sparsity conditions familiar from compressed sensing, can in principle support uniqueness, it remains unclear whether the mixing matrix **A** in our setting satisfies the specific sparsity or incoherence conditions required. Exploring these assumptions is an interesting direction for future work.

## N    VERIFYING EXPONENTIAL SUBSTITUTES ON PYTHIA-70M

A core modeling choice in ConCA is the replacement of the exponential function with numerically stable surrogates such as SoftPlus, motivated by the fact that LLM activations lie in ranges that make the true exponential function prone to gradient explosion and instability. To further validate this design choice, we conduct an additional controlled experiment where the exponential function becomes numerically stable.

Specifically, we construct a toy setting by clamping the final-layer hidden activations of Pythia-70m to bounded ranges where exponential function can be safely evaluated without numerical overflow. We consider the four clamping windows as shown in Table 4. We additionally include LayerNorm and the same sparsity penalty used in our SoftPlus variant to ensure comparability. We then train ConCA variants using the true exponential under these artificially stabilized conditions, and compare them with our SoftPlus-based variant. Performance is evaluated using both Pearson correlation (between recovered concepts and linear-probe ground truth) and MSE reconstruction error.

Three observations emerge: Exp becomes usable only under artificial clamping. Even when stabilized, exp-based ConCA exhibits lower Pearson correlation ( 0.69–0.72) and higher MSE ( 10–12) across all clamp settings. SoftPlus consistently outperforms exp, achieving both the highest correlation ( 0.73) and the lowest MSE ( 7–8), despite being evaluated under more challenging, unclamped activation distributions. Together, these findings demonstrate that the surrogate activations used in ConCA are not only numerically safer but also better aligned with empirical behavior, even in settings explicitly constructed to favor the exponential function. This experiment thus reinforces our modeling choice and supports the theoretical motivation behind replacing exp in ConCA.

## O    ACTIVATION PATCHING EVALUATION

Activation patching is widely regarded as one of the strongest tests of functional interpretability. Unlike reconstruction-based metrics (e.g., MSE) or alignment metrics (e.g., Pearson correlation), activation patching directly measures whether a representation—after being encoded and reconstructed by a concept-extraction model—still leads the LLM to make similar token-level predictions. To evaluate this, we measure how substituting an internal hidden state with its ConCA/SAE reconstruction affects the LLM's output logits. If a dictionary model faithfully captures the underlying functional structure, the LLM's predictions should remain largely unchanged.

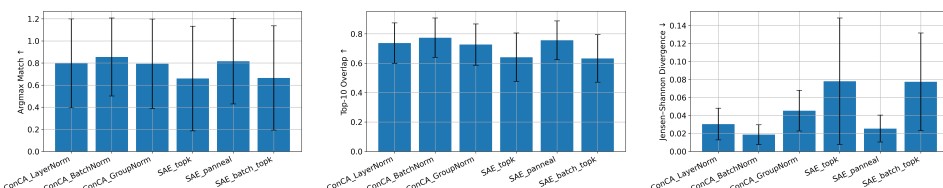

Figure 17: Activation patching comparison across ConCA and SAE variants on Pythia-70M using 10,000 cached hidden activations from The Pile. (a) Argmax Match (higher is better): ConCA variants preserve the model's top prediction more reliably than SAE baselines. (b) Top-10 Overlap (higher is better): ConCA reconstructions retain substantially more of the model's predictive structure. (c) Jensen–Shannon Divergence (lower is better): ConCA achieves lower distributional distortion, indicating higher functional faithfulness.

We use 10k randomly sampled token activations from The Pile activation dataset of Pythia-70m. Sampling is performed once, cached, and reused across models to ensure perfect comparability. For each sampled activation, we: 1: feed the original hidden state into the model's final prediction layer to obtain the baseline logits. 2: reconstruct the hidden state using either our ConCA or SAEs. 3: feed the reconstructed activation back into the model and compare the resulting logits with the baseline. We report three widely used functional metrics: Argmax Match: whether the top predicted token stays the same. Top-10 Overlap: fraction of overlapping tokens in the top-10 predictions. Jensen–Shannon Divergence: distributional distance between original and reconstructed logits (lower is better). Each metric is averaged over all 10,000 activations.

The results in Figure 17 show that ConCA-BatchNorm achieves the best overall performance across metrics. SAE-top-k and SAE-batch-top-k show the largest divergence, indicating functional mismatch despite good sparsity. ConCA variants exhibit lower variance, suggesting more stable behavior across diverse activations. This experiment demonstrates that: ConCA reconstructions lead the LLM to make more similar predictions, confirming that ConCA retains functional information more faithfully than SAEs.

# P  EXPERIMENTS ON ADDITIONAL COUNTERFACTUAL CONCEPT PAIRS

While the original 27 counterfactual concept pairs from Park et al. (2023) provide clean, expert-curated evaluations along several core linguistic axes (verb inflections, adjective morphology, noun attributes, and multilingual translations), they cover only a small portion of the semantic space relevant to modern LLM behavior. In particular, many practically important concepts, such as sentiment polarity, toxicity, factuality, stance, politeness, or degree/intensity—are not represented in the original benchmark. These dimensions are widely studied in interpretability and safety research, and their inclusion offers a more comprehensive evaluation of concept alignment.

To complement the original dataset, we construct 23 additional counterfactual concept pairs, each designed to capture a single semantic concept for linear probing and our theoretical disentanglement analysis. For every concept, we curate 50 pairs where only the targeted semantic attribute changes while all other factors (POS category, morphological structure, lexical frequency, etc.) remain controlled.

These new concepts span five broader semantic families—sentiment polarity, toxicity/politeness, factuality/truthfulness, stance/subjectivity, and degree/intensity—and are summarized in Table 5. Importantly, the pairs were constructed to satisfy the same identifiability constraints emphasized in our main text: each pair differs in one concept, preventing confounding correlations across multiple latent factors. This makes the dataset suitable both for linear probing and for measuring concept-level disentanglement.

A concise overview of the 23 added concepts is provided (full list and examples in Table 5). Overall, the results in the left in Figure 18 show that the ConCA variants demonstrate substantially stronger alignment than standard SAE baselines. In Pythia-70M, ConCA methods outperform SAEs by a large margin, indicating more faithful feature extraction in low-capacity language models. As model size increases to 1.4B and 2.8B, all methods improve, but ConCA retains a consistent advantage: LayerNorm-ConCA yields the highest and most stable correlations across scales, followed closely

Table 5: Additional counterfactual concept pairs (sentiment, toxicity, factuality/truthfulness, stance, politeness). Each concept contains 50 pairs.

| # | Concept | Example Pair | Word Pair Counts |
|---|---|---|---|
| *Sentiment polarity* | | | |
| 1 | positive → negative | (happy, sad) | 50 |
| 2 | positive → neutral | (amazing, average) | 50 |
| 3 | negative → neutral | (terrible, ordinary) | 50 |
| *Toxicity* | | | |
| 4 | toxic → neutral | (stupid, silly) | 50 |
| 5 | toxic → polite | (idiot, friend) | 50 |
| 6 | rude → polite | (shut up, please speak) | 50 |
| *Factuality / Truthfulness* | | | |
| 7 | true → false | (earth, flat-earth) | 50 |
| 8 | factual → nonfactual | (oxygen, magic-power) | 50 |
| 9 | real → fictional | (doctor, wizard) | 50 |
| *Stance / Subjectivity* | | | |
| 10 | supportive → opposed | (agree, oppose) | 50 |
| 11 | approving → disapproving | (praise, blame) | 50 |
| 12 | subjective → objective | (biased, neutral) | 50 |
| *Politeness / Formality* | | | |
| 13 | polite → impolite | (sorry, shut-up) | 50 |
| 14 | formal → informal | (assist, help) | 50 |
| 15 | respectful → disrespectful | (sir, dude) | 50 |
| *Emotion / Tone* | | | |
| 16 | calm → angry | (calm, furious) | 50 |
| 17 | excited → bored | (excited, uninterested) | 50 |
| 18 | friendly → hostile | (friendly, hostile) | 50 |
| *Intensity / Degree* | | | |
| 19 | mild → strong | (warm, hot) | 50 |
| 20 | weak → strong | (soft, solid) | 50 |
| 21 | low-certainty → high-certainty | (maybe, definitely) | 50 |
| *Common semantic axes* | | | |
| 22 | general → specific | (animal, dog) | 50 |
| 23 | concrete → abstract | (chair, justice) | 50 |

by GroupNorm and BatchNorm. In contrast, SAE-Top-k and Batch-Top-k lag significantly behind, and Panneal improves with scale but remains below ConCA. These results indicate that ConCA's normalization-driven design leads to more stable and interpretable feature learning, particularly in smaller models where dictionary training is more brittle.

## Q    DETAILED RESULTS OF FIGURE 3

While Figure 3 includes error bands reflecting variability across random seeds, many of these standard deviations are extremely small and therefore not visually distinguishable in the plots. To make these differences explicit, we report the full numerical mean and std values for each method–model pair in Tables 6 and 7.

## R    RELATION WITH LINEAR REPRESENTATION HYPOTHESIS

In Theorem 2.1, the mixing matrix $\mathbf{A}$ offers a way to interpret how latent semantic information may be organized within the representation space of LLMs. At a high level, $\mathbf{A}$ can be viewed as describing

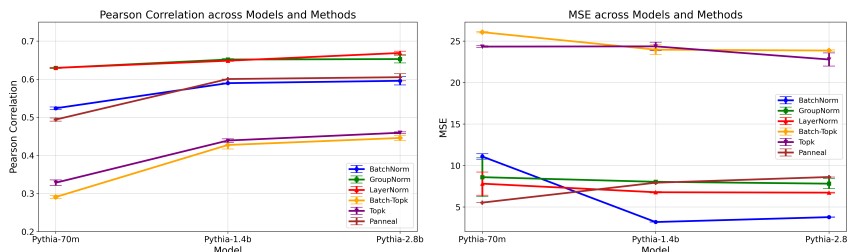

Figure 18: Pearson correlation and MSE between predicted concept logits and matched dictionary features across model scales (Pythia-70M, 1.4B, 2.8B) and dictionary learning methods. Mean ± standard deviation over three random seeds is shown. ConCA variants achieve higher Pearson correlation than SAE baselines, with the performance gap most visible in smaller models and remaining stable as scale increases.

| Method | Model | Pearson (MPC) | MSE |
|---|---|---|---|
| BatchNorm | pythia-70m | $0.6500 \pm 0.00552$ | $8.3305 \pm 4.10287$ |
| | pythia-1.4b | $0.7543 \pm 0.00004$ | $1.6418 \pm 0.00135$ |
| | pythia-2.8b | $0.7630 \pm 0.00132$ | $2.4657 \pm 0.04794$ |
| GroupNorm | pythia-70m | $0.7285 \pm 0.00094$ | $9.0071 \pm 1.36961$ |
| | pythia-1.4b | $0.7914 \pm 0.00222$ | $6.3021 \pm 0.09493$ |
| | pythia-2.8b | $0.8038 \pm 0.00059$ | $7.1652 \pm 0.30648$ |
| LayerNorm | pythia-70m | $0.7325 \pm 0.00015$ | $7.6272 \pm 0.40461$ |
| | pythia-1.4b | $0.7958 \pm 0.00009$ | $5.2756 \pm 0.09484$ |
| | pythia-2.8b | $0.8074 \pm 0.00178$ | $3.5708 \pm 0.45880$ |
| Batch-TopK | pythia-70m | $0.5687 \pm 0.00691$ | $24.6165 \pm 0.10317$ |
| | pythia-1.4b | $0.6856 \pm 0.00205$ | $24.3850 \pm 0.08773$ |
| | pythia-2.8b | $0.7037 \pm 0.00211$ | $23.7690 \pm 0.07034$ |
| TopK | pythia-70m | $0.5851 \pm 0.01090$ | $22.9035 \pm 0.03389$ |
| | pythia-1.4b | $0.6969 \pm 0.00399$ | $22.7108 \pm 0.10553$ |
| | pythia-2.8b | $0.7074 \pm 0.00810$ | $22.5243 \pm 0.03869$ |
| Panneal | pythia-70m | $0.6474 \pm 0.00816$ | $7.7097 \pm 0.08165$ |
| | pythia-1.4b | $0.7413 \pm 0.00234$ | $5.7218 \pm 0.08165$ |
| | pythia-2.8b | $0.7613 \pm 0.00205$ | $6.3563 \pm 0.03300$ |

Table 6: Comparison of Pearson correlation (MPC) and MSE across normalization methods and Pythia model scales. Each cell reports mean ± standard deviation over random seeds.

how the log-posteriors of latent concepts might be linearly combined inside hidden activations. This perspective is related to the broader *Linear Representation Hypothesis*, which suggests that certain semantic attributes in neural representations may interact in an approximately linear manner.

Under this view, each column of $\mathbf{A}$ may be interpreted as indicating a possible direction associated with a particular latent factor, while differences across rows could correspond to the kinds of *difference vectors* often observed in counterfactual pairs or steering-vector analyses. In particular, when two inputs differ only in one latent concept, their representation difference may align with the corresponding column of $\mathbf{A}$, reminiscent of the vector arithmetic phenomena discussed in both word embeddings and modern LLMs.

Such observations hint that $\mathbf{A}$ may induce a geometric structure in which examples sharing similar latent concept values tend to cluster along certain affine subspaces, while changes in concept values correspond to movement along interpretable directions. From this perspective, the mixing matrix provides a possible explanation for why linear unmixing methods like ConCA can recover meaningful concept-level variations in practice. Of course, these interpretations are exploratory, and the precise geometric structure may depend on various factors such as model architecture, data distribution, and training objective.

| Method | Model | Pearson (MPC) | MSE |
|---|---|---|---|
| BatchNorm | Qwen3-1.7B | $0.7281 \pm 0.00050$ | $3.3193 \pm 0.00026$ |
| | gemma-3-1b-pt | $0.6853 \pm 0.00374$ | $1.5604 \pm 0.00057$ |
| | pythia-1.4b | $0.7543 \pm 0.00004$ | $1.6418 \pm 0.00135$ |
| GroupNorm | Qwen3-1.7B | $0.7979 \pm 0.00023$ | $6.1347 \pm 0.00051$ |
| | gemma-3-1b-pt | $0.7791 \pm 0.00123$ | $6.5240 \pm 0.05917$ |
| | pythia-1.4b | $0.7914 \pm 0.00222$ | $6.3021 \pm 0.09493$ |
| LayerNorm | Qwen3-1.7B | $0.7900 \pm 0.00034$ | $5.9607 \pm 0.05288$ |
| | gemma-3-1b-pt | $0.7738 \pm 0.00019$ | $6.1671 \pm 0.12262$ |
| | pythia-1.4b | $0.7958 \pm 0.00009$ | $5.2756 \pm 0.09484$ |
| Batch-TopK | Qwen3-1.7B | $0.6926 \pm 0.01003$ | $18.1650 \pm 0.11887$ |
| | gemma-3-1b-pt | $0.6617 \pm 0.00301$ | $44.4503 \pm 0.20351$ |
| | pythia-1.4b | $0.6856 \pm 0.00205$ | $24.3850 \pm 0.08773$ |
| Panneal | Qwen3-1.7B | $0.7345 \pm 0.00500$ | $3.1175 \pm 0.03114$ |
| | gemma-3-1b-pt | $0.6770 \pm 0.00419$ | $2.7399 \pm 0.02055$ |
| | pythia-1.4b | $0.7413 \pm 0.00234$ | $5.7218 \pm 0.08165$ |
| TopK | Qwen3-1.7B | $0.7027 \pm 0.00024$ | $15.0734 \pm 0.03822$ |
| | gemma-3-1b-pt | $0.6758 \pm 0.00334$ | $42.6148 \pm 0.18811$ |
| | pythia-1.4b | $0.6969 \pm 0.00399$ | $22.7108 \pm 0.10553$ |

Table 7: Comparison of Pearson correlation (MPC) and MSE across methods and architectures. Values are mean ± standard deviation over random seeds.

