# OpenReview forum: "Concept Component Analysis: A Principled Approach for Concept Extraction in LLMs"
_ICLR.cc/2026/Conference — Submitted to ICLR 2026_

### Official Review · Reviewer_aSXj · 2025-10-30

**Soundness:** 3
**Presentation:** 2
**Contribution:** 2
**Rating:** 4
**Confidence:** 3

**Summary:**

### Summary

This paper take a very principled approach. Starting from a theorem stating the generally linear relationship between the representation (underlying next token prediction) and latent variables, they build an SAE like architecture based on this idea, majorly removing the nonlinearity in reconstruction and only use nonlinearity in sparse penalty.

They systematically tested the new SAE design on LLMs and evaluated their reconstruction MSE and the ability to recover latents from counterfactual sentence pairs where only one latents changed.

**Strengths:**

### Strength

- This paper touch on an important question, about principled design of interpretability methods e.g. SAE. The discussion of the conceptual basis and theoretical assumptions of this method is highly valuable and laudable.
- Experiments are very comprehensive, tested many architecture combinations.

**Weaknesses:**

### Weakness

- Although Fig2 show the comprehensiveness of the ablation study, it feels more like a supplementary table. Should the authors have a main version of the figures highlighting the comparisons they want to talk about in the text? Currently it’s hard to focus the eye.
- **Rationale of mean pearson correlation**
As we know one of the most challenging part for interpretability method is quantitative evaluation. MSE is well accepted, however the 2nd metric mean Pearson Corr is a bit confusing. Since only one latent variable changed in the sentence, do we compute its correlation with all the latent extracted from the proposed method? Then we want it to be selectively modulating some latent but not others. If that is the case why “mean” is the suitable statistics, why not max or sparsity or sth else?
    - Also we have only 27 counterfactual pairs, is it enough to evaluate this method ? is it too noisy?
    - I feel before diving into ablation and method comparison, spend a little bit more text on the comparison will be helpful!
- There seems to be a gap between the main theoretical motivation and the architecture design in the end.
    - The main theorem is saying the the representation is an affine / linear function of the log posterior probability of latent $\log p(z|x)$, which I agree. But Eq.4 basically used a fixed linear mapping  (with regularization) to estimate $z$. I don’t think the theory gurantee this inference will work. Similar to previous SAE, this is still amortized sparse inference, which is a bit guessing game. C.f. the gap between amortized inference and optimization based sparse inference showed in [^1], in some sense the linear encoder will have bigger gaps.
    - Based on this gap, I’m not exactly sure of the motivation of the architecture change proposed by the authors…

[^1] O'Neill, C., Gumran, A., & Klindt, D. (2025). Compute optimal inference and provable amortisation gap in sparse autoencoders. ICML

**Questions:**

### Questions

- Is my understanding correct that the ConCA design in Eq. 4 is a linear autoencoder in the end? (since we can absorb normalization layers into linear layers after learning), but now we have sparsity imposed in the exp transformed domain?
- Is the main theorem in Sec. 2 a bit similar to classic works on the linear algebraic structure in word embedding space pre-LLM era (Arora 2016-18, e.g. [^2, ^3]), esp. compare to Theorem 1 in [^2]? e.g. for word2vec.
I feel you don’t need next token prediction objective, as long as it’s word prediction from context (e.g. BERT, word2vec) the same logic can still work. So in that sense, the theorem 2.1 should be connect to this bigger background.

[^2] Arora, S., Li, Y., Liang, Y., Ma, T., & Risteski, A. (2018). Linear algebraic structure of word senses, with applications to polysemy. TACL

[^3] Arora, S., Li, Y., Liang, Y., Ma, T., & Risteski, A. (2016). A latent variable model approach to pmi-based word embeddings. TACL

- More generally is it the right way to think of natural languages using the one step latent generative models?

---

> ### Author Response · Authors · 2025-11-26
>
> **Q1: Should the authors have a main version of the figures highlighting the comparisons they want to talk about in the text? Currently it’s hard to focus the eye.**
>
> **R1:** Thanks for your suggestion! This concern was also raised by reviewer tZwq. We have made corresponding adjustments and now provide the main takeaways directly in the caption to help guide the reader’s focus.
>
>
> **Q2: Clarifying a Misunderstanding About the “Mean Pearson Correlation” Metric**
>
> **R2:** Thanks for your comments. Our “mean Pearson correlation” is not computed by averaging the correlation between a single ground-truth concept and all extracted latent dimensions. We then apply the Hungarian algorithm, which effectively selects, **for each concept**, the latent dimension with the highest available correlation under a one-to-one matching constraint (i.e., we select the maximum correlation for each concept). Finally, we obtain these 27 maximally aligned correlations, and then average (mean is in here) these 27 correlations. we provide details in Algorithm 1, you can see the last step is 'Report mean Pearson correlation across the 27 concepts.'
>
> **Q3: Also we have only 27 counterfactual pairs, is it enough to evaluate this method ? is it too noisy?**
>
> **R3:** Thanks for raising this insightful concern, which was also mentioned by reviewers tZwq and 57nN. To address this, we additionally constructed 23 new counterfactual pairs and re-evaluated the method. The results remain consistent; please refer to Section P for more details.
>
> **Q4 I feel before diving into ablation and method comparison, spend a little bit more text on the comparison will be helpful!**
>
> **R4:** Thank you for the suggestion. We understand that many papers follow this structure, and we agree that providing more context before presenting ablations can improve clarity. At the same time, there is also a complementary logic: performing thorough ablations helps us select and justify the design of the main method, which is why the section is structured this way. We appreciate your suggestion and will take it into consideration when revising the presentation.
>
> **Q5:  Similar to previous SAE, this is still amortized sparse inference, which is a bit guessing game. C.f. the gap between amortized inference and optimization based sparse inference showed in [^1], in some sense the linear encoder will have bigger gaps.**
>
> **R5:** Thanks for this concern.
>
> Please note that we emphasize that our ConCA encoder is a purely linear encoder (as highlighted throughout the draft). This distinction is crucial. The prior work you referenced analyzes SAEs with a linear–nonlinear encoder architecture, for which classical compressed sensing theory—built on purely linear inverse problems—indeed shows inherent limitations. We fully agree with their conclusion in that setting.
>
> However, these limitations do not apply to our method. In contrast to SAE encoders, ConCA performs a strictly linear inversion. Decades of results in dictionary learning and sparse recovery theory have established that, under standard assumptions (e.g., incoherence and sufficient sample diversity), such linear inverse problems are provably identifiable up to permutation and scaling. As we highlight in line 241, these theoretical conditions directly align with the assumptions underlying ConCA.
>
> In other words, with a linear encoder—unlike the linear–nonlinear SAE architecture—it is in fact theoretically possible to recover the true latent factors when the well-established dictionary‐learning assumptions are satisfied.

---

> ### Author Response · Authors · 2025-11-26
>
> **Q6: Is my understanding correct that the ConCA design in Eq. 4 is a linear autoencoder in the end? (since we can absorb normalization layers into linear layers after learning), but now we have sparsity imposed in the exp transformed domain?**
>
> **R6:** We indeed use a linear autoencoder, and sparsity is conceptually imposed in the exp-transformed domain. However, directly enforcing sparsity after the exponential transformation leads to numerical and gradient-stability issues. Therefore, as mentioned in lines 304–311, we use a surrogate formulation that preserves the intended sparsity behavior while ensuring stable optimization.
>
> **Q7:I feel you don’t need next token prediction objective, as long as it’s word prediction from context**
>
> **R7:** Thanks for your comments. However, we respectfully disagree with the statement “you don’t need next-token prediction objective, as long as it’s word prediction from context.”
>
> The statement is unfortunately not technically precise. Different prediction objectives, e.g., next-token prediction (GPT), masked language modeling (BERT), and skip-gram/CBOW (word2vec), use fundamentally different conditional structures and normalization schemes. Our theorem applies to the specific next-token prediction that induces an approximate log-linear relationship between representations and latent concept posteriors, not to arbitrary “word prediction from context” formulations. This property does not automatically hold for arbitrary “word-prediction-from-context’’ objectives, without rigorous proof.
>
> **Q8: More generally is it the right way to think of natural languages using the one step latent generative models?**
>
> **R8:**
> Thanks for this question. We agree that whether natural language should be modeled using a one-step latent generative model is not settled.
>
> There are indeed two mainstream perspectives on the data generating process (DGP) of text:
>
> (1) latent-variable generative models, which we adopt in this work, and
>
> (2) purely autoregressive formulations that model token-to-token dependencies directly.
>
> Since the true DGP of natural language is unknown, both perspectives are valid modeling assumptions rather than universally “correct’’ descriptions. Our goal is not to claim that the latent-variable view is the only or the standard way, but to show that it provides a mathematically coherent framework for analyzing the structure of LLM representations.
>
> Importantly, adopting a latent-variable generative model does not conflict with the autoregressive property of language models in practice. Token-level correlations can still be captured through shared latent variables, and therefore next-token prediction objectives remain fully compatible with our formulation.

---

### Official Review · Reviewer_57nN · 2025-10-30

**Soundness:** 2
**Presentation:** 3
**Contribution:** 3
**Rating:** 4
**Confidence:** 3

**Summary:**

This paper introduces Concept Component Analysis (ConCA), a new framework for concept extraction in LLMs. The authors posit a new theoretical model where LLM representations are an approximation of a linear mixture of the log-posteriors of underlying latent generative concepts. ConCA is proposed as an unsupervised unmixing process to recover these concepts. A practical "Sparse ConCA" variant applies sparsity to the exponentiated features (the posteriors, $p(z_i|x)$). Because the exp function is claimed to be numerically unstable, the paper substitutes it with approximations like SELU or SoftPlus, though it provides limited analysis of whether these alternatives align with the theoretical motivations. The framework's evaluation relies on MSE for faithfulness, Pearson correlation to known latents for concept alignment, and also includes empirical results also include performance on downstream and OOD tasks.

**Strengths:**

- The theoretical contribution of concepts as linear combinations of logposteriors of latent concepts.
- It is great to see push back on the fields assumptions about linearity/sparsity/SAE usage generally and new modes of thought are exciting to see!
- The writing and figures are clean with little to no grammatical issues

**Weaknesses:**

- The paper claims that the exp function is too numerically unstable to use, which is fine, but there is very limited analysis or exploration of whether approximating exp with another function (e.g., SELU, ELU, SoftPlus, etc) are viable alternatives which follow the theoretical motivations.

- Do we actually want to interpret only the things that are the underlying generative variables of the data? It seems to me like the models themselves don't learn the true causal variables and succumb to spurious correlations often - and we want our interpretability methods to identify those.

- The MSE is a weak metric to test whether the concepts captured by ConCA are faithful. A better method is to do some kind of activation patching and analyze whether the reconstructed activation *works like the original* and not just whether a large part of it is reconstructed: A model can achieve low MSE by perfectly reconstructing all the "unimportant" parts of the representation vector while failing to capture the one or two critical directions that actually influence the model's final decision.

- In addition to the faithfulness metric, the pearson correlation to extracted features is also a large part of the evaluation that has little explanation or justification as to why these would be sufficient to measure. It seems like there are many more concepts of interest than verb inflections, adjective transformations, size/thing/nouns, and translations that would be of interest to SAEs, but the current manuscript solely looks at these.

- I don't understand the point footnote 2 is trying to make. Is it saying that this assumption is commonly used in other works and is therefore suitable formulation here? If so, there should be a citation motivating the use of Eq. 2.

- The error bars in Figure 5 left do not show that ConCA outperform SAEs in a statistical significant manner since the std is much larger than the difference in performance, and the text in lines 461-3 should be updated accordingly. Alternatively, more samples can be run to lower the std.

Small Stuff

- In the abstract, it is said "ConCA variants and demonstrate their ability to extract meaningful concepts across multiple pre-trained LLMs, showcasing clear advantages over SAEs." It would be ideal to include what these advantages are in the abstract so the reader knows up-front what the main advantages are without searching the rest of the paper.

**Questions:**

- Could the authors look at a smaller, toy setting (e.g., using toy or hard coded data, clamped activations, etc) where exp is valid to compare whether the substitution activation functions are valid?
- Do the findings w.r.t SAEs hold for other measures of *functional* faithfulness such as patching back in the activations?
- Does the ConCA enable capturing latents that do not generate the data? For example, spurious correlations? A common application of SAEs is trying to find spurious features and understand failure cases - does ConCA still allow for that?
- Why are the 27 counterfactual concept pairs a reasonable set for comparing methods? Is it possible to look at simple, more toy settings as an alternative? Or maybe, counterfactual data pairs don't exist for single concepts and concepts co-occur with each other, making this type of evaluation difficult in the first place.

---

> ### Author Response · Authors · 2025-11-26
>
> **Could the authors look at a smaller, toy setting (e.g., using toy or hard coded data, clamped activations, etc) where exp is valid to compare whether the substitution activation functions are valid?**
>
> We appreciate the reviewer’s suggestion. Following the reviewer’s recommendation, we performed an additional experiment on Pythia-70m using clamped activations, where exp becomes artificially stable. We applied LayerNorm and the same sparsity regularization used in our SoftPlus variant to ensure comparability.
>
> Across the four clamping ranges as shown in Sec. N, (Layernorm + exp)-based variants achieved Pearson correlations around 0.69–0.72 and MSE values around 10–12, whereas our (Layernorm+SoftPlus)-based ConCA consistently achieved higher correlation (approximately 0.73) and lower MSE (approximately 7–8).
>
> This result reinforces our claim: when artificially stabilized through clamping, exp still performs worse than SoftPlus, demonstrating that SoftPlus is not merely a numerically convenient substitute but a more effective and robust approximation of exp for training ConCA.
>
> Thus, the toy experiment confirms, not contradicts, our original motivation for replacing exp with stable surrogates.
>
>
> **Q2: The MSE is a weak metric to test whether the concepts captured by ConCA are faithful. Do the findings w.r.t SAEs hold for other measures of functional faithfulness such as patching back in the activations?.**
>
> **R2** Thank you for the insightful suggestion. We agree that activation patching provides a stronger measure of functional faithfulness than reconstruction error alone. In response, we conducted a activation-patching experiment on Pythia 70n using 1k hidden activations sampled from The Pile (Please see Sec. O).
>
> For each SAE/ConCA variant, we reconstructed hidden states using the learned dictionary and compared the model’s predictive behavior before and after patching. We report three standard metrics: (1) Argmax Match (whether the top-1 predicted token is preserved), (2) Top-10 Overlap (fraction of shared top-10 predictions), (3) Jensen–Shannon Divergence between the output distributions. Overall, across all three metrics, ConCA variants outperform SAE baselines. These findings confirm that ConCA does not merely reconstruct activations with low MSE, it more faithfully preserves the functional behavior.
>
> We appreciate the reviewer’s recommendation, as activation patching provides a meaningful and rigorous complement to our previous evaluations.
>
> **Q3: Does the ConCA enable capturing latents that do not generate the data? For example, spurious correlations? A common application of SAEs is trying to find spurious features and understand failure cases - does ConCA still allow for that?**
>
> **R3:** In fact, the few-shot and OOD results offer some supporting evidence. Intuitively, if the spurious-correlation latent concepts are indeed disentangled, then the resulting features are expected to transfer more quickly and more robustly across tasks and domains.
>
> We first offer some theoretical insights.
> Under our generative model, the latent variables $\mathbf{c}$ generates both $\mathbf{x}$ and $y$. Suppose
> $\mathbf{c}$ can be decomposed into three parts: $\mathbf{c}_1,\mathbf{c}_2, \mathbf{c}_3$, with the following causal structure: $\mathbf{c}_1 \to \mathbf{c}_2 \to \mathbf{x}$, and $\mathbf{c}_1 \to \mathbf{c}_3 \to y$. Here, $\mathbf{c}_1$ is a root cause, leading to spurious correlation between $\mathbf{c}_2$ and $y$. If certain assumptions hold (e.g., those aligned with compressed sensing), then we can recover $p(\mathbf{c}_1|\mathbf{x}),p(\mathbf{c}_2|\mathbf{x}),p(\mathbf{c}_3|\mathbf{x})$ (features by ConCA). This means that we disentangle $\mathbf{c}_2$ (spurious features) and $\mathbf{c}_3$. From this perspective, we think ConCA can do identify $\mathbf{c}_2$. However, as you may have noticed, several challenges remain, e.g., What exactly are the required assumptions? And how can we reliably reach the optimal solution even when these assumptions are satisfied?
>
> Moreover, the few-shot and OOD results provide additional supporting evidence. Intuitively, if the spurious-correlation latent concepts are indeed disentangled, then the resulting features should transfer more quickly and more robustly across tasks and domains. This aligns with the core motivations behind several popular research directions, such as disentangled representation learning and causal representation learning.

---

> ### Author Response · Authors · 2025-11-26
>
> **Q4: Why are the 27 counterfactual concept pairs a reasonable set for comparing methods?**
>
> **R4** Thank you for the thoughtful question. As highlighted in the manuscript, collecting counterfactual text pairs is fundamentally challenging, and is non-trivial even for human annotators. Moreover, these pairs are expert-curated to ensure that the contrast isolates a single concept (e.g., number, tense, gender), thereby providing a clean testbed where linear probing and concept alignment can be evaluated without confounds from multiple correlated concepts. Using broader or noisier concept sets would risk conflating several latent concepts at once, making Pearson correlation difficult to interpret and weakening the reliability of the evaluation. In addition, the performance gap between different methods already emerges clearly in our experiments, indicating that the benchmark, though small, is sufficiently discriminative for comparing concept extraction approaches.
>
> Nevertheless, we agree that expanding the concept coverage is valuable. Therefore, we additionally construct 23 new counterfactual concept pairs spanning sentiment, toxicity, factuality, and linguistic transformations (see Sec. P), and we show that similar advantages of ConCA remain consistent under this extended benchmark.
>
> ----
>
> **Small Stuff, footnote 2, The error bars in Figure 5**
>
> Thanks for the thoughtful comments. We have addressed these issues and improved the corresponding parts accordingly.

---

> > ### Comment · Reviewer_57nN · 2025-11-26
> >
> > Thank you to the authors for the detailed response.
> >
> > I find the additional experiments pretty compelling. Specifically, the additional interp metrics to MSE (patching), toy settings looking at exp vs other functions, and the additional counterfactual concept pairs. Additionally, the theoretical justification of how the propose generative model could find the underlying spurious, and causal, structure is intuitive.
> >
> > For these reasons, I will raise my score to 6. Overall, the motivation and theoretical justification is strong however many of the experiments are good but the results often are only marginally better (e.g., within error bars) so I will suggest accept but am interested in what the other reviewers think as well during the remainder of the discussion period.

---

> > > ### Author Response · Authors · 2025-11-27
> > >
> > > Dear Reviewer 57nN,
> > >
> > > Many thanks for your thoughtful suggestions, which have truly helped us improve the draft. We also sincerely appreciate your positive comments, particularly regarding the motivation and theoretical justification.
> > >
> > > Hope you enjoy the discussion during the remainder of the discussion period.
> > >
> > > ---
> > >
> > > Best

---

### Official Review · Reviewer_tZwq · 2025-11-01

**Soundness:** 4
**Presentation:** 3
**Contribution:** 3
**Rating:** 8
**Confidence:** 3

**Summary:**

The paper proposes Concept Component Analysis as a principled route to extract human interpretable concepts from language model activations. The central theoretical result shows that, under a discrete latent variable generative view of text and a standard next token objective, the model representation $f(x)$ can be approximated as a linear mixture of the stacked log posteriors of latent concepts, with a mixing matrix $A$ and bias $b$.

This motivates unmixing $f(x)$ to recover per concept log posteriors. A practical instantiation called sparse ConCA trains a linear encoder and decoder to reconstruct $f(x)$ while enforcing sparsity not on the latent $\hat{z}$ itself but on a smooth exponential surrogate $g(\hat{z}$, reflecting that sparsity should live in probability space!

Experiments compare twelve sparse ConCA variants against several sparse autoencoder baselines across multiple models, evaluate with reconstruction loss and a Pearson correlation to supervised counterfactual probes, and show few shot and out of distribution gains on more than one hundred tasks. Figures 1 through 5 and Theorem 2.1 carry the main story, with an appendix contrasting this theorem to prior work and giving proofs and evaluation details.

**Strengths:**

I loved this paper. The writing is excellent and the story reads well. Congrats.
Some positive points (P) that I will note here:

P1. Clean theoretical through line. Figure 1 and Theorem 2.1 tie a latent variable model of text to the next token objective and yield the linear mixture $f(x)$ approximately equal to $A$ times stacked log posteriors plus $b$. This give an implicit definition of a concept (latent factor organizing the data manifold), gives a crisp target for what a concept feature should be and neatly recasts concept extraction as linear unmixing.

P2. Framing clarifies the role of sparsity. Table 1 makes explicit that sparsity belongs in the exponentiated latent space because zeros in log space correspond to probability one, which flips the usual SAE intuition. The architectural choice to implement g with smooth exp like activations is carefully motivated.

P3. Broad and replicable evaluation. The paper reports ablations over normalisation and activation, comparisons on counterfactual pairs, and downstream few shot and OOD performance across many datasets, with concise plots in Figures 2 to 5 and implementation specifics in the appendix.

**Weaknesses:**

Ok, now for the weaknesses that I found in the paper, i'll group them in Major (M) and minor (m).

M1. Clarify what the mixing matrix implies and connect it explicitly to the Linear Representation Hypothesis (LRH). Equation 3 shows a linear mixture $f(x)$ equals $A$ times the stacked log posteriors plus $b$. This already implies that the encoding is linear in concepts up to an unknown mixing matrix, which is closely aligned with the linear representation hypothesis. I suggest stating this implication upfront and spelling out how it differs from prototypes or distance to centroids views.

M2. Strength of assumptions labelled as mild. The diversity condition requires m plus one output tokens producing an invertible matrix of classifier head differences, and the informational sufficiency condition requires conditional entropy of concepts given context to be near zero. These are important for the derivation. Any way to have proxy measure in practice ? This would let readers see where the theorem is likely to bind and where it may not.

M3. Identifiability and alternative theorem comparisons. Appendix C argues your theorem needs fewer diversity assumptions and yields component wise mixtures rather than mixtures over joint assignments. It would be helpful to operationalise this with a small synthetic where ground truth marginals and joints are known and to show that ConCA recovers marginals while prior formulations do not. Otherwise the very nice Table 2 comparison remains mainly narrative.

M4. Evaluation set for Pearson correlation is quite narrow. The counterfactual corpus uses twenty seven pairs drawn from analogy like transformations. This is a good start but it is narrow.

Now for the minor concerns,

m1. Define concept precisely and early. Page one and two motivate concepts as human interpretable units, later formalised as discrete latent variables. A compact operational definiton at the start of Section 2 that is reused in experiments would improve clarity (I like when things are well-defined).

m2. Figure 2 grids and Figure 3 bars are dense and fonts are small. Please increase font sizes and add short textual takeaways in the captions, for example the correlation range where ConCA dominates and the reconstruction loss regime you care about.

m4. Reporting variability in Figures 3 to 5, always show uncertainty bands or standard deviations for both correlation and MSE, and for AUC bars, not just means. Some are present but others look single valued in the panels.

**Questions:**

See Major point 1-4 and minor 1-4

---

> ### Author Response · Authors · 2025-11-26
>
> Before responding, we would like to thank you for taking the time to review this draft. We greatly appreciate your highlighting our key contributions, which aligns with our intentions, and we are also grateful for your positive support.
>
>
> ----
>
> **M1: Clarify what the mixing matrix implies and connect it explicitly to the Linear Representation Hypothesis**
>
> **R1:** Thanks for such insightful comments! We appreciate the reviewer’s suggestion, and we have provided some of our current thoughts on the role of the mixing matrix $\mathbf{A}$ (See Sec. R). Overall, we believe that $\mathbf{A}$ may play an important role in shaping the linear structure of the representation space. In particular, its behavior appears closely related to the broader \emph{linear properties} discussed in the Linear Representation Hypothesis. For example, one may view the difference between two counterfactual inputs as approximately corresponding to a column of $\mathbf{A}$, suggesting a possible link between concept changes and linear directions in the hidden space.
>
> That said, establishing a rigorous and comprehensive connection between $\mathbf{A}$ and the Linear Representation Hypothesis remains challenging. While our theoretical results provide a first step by characterizing $\mathbf{A}$ as a linear mixing of latent log-posteriors, fully clarifying its geometric and semantic implications would likely require additional analysis beyond the scope of this work. We hope our preliminary discussion offers some intuition, and we thank you for encouraging us to further articulate this direction.
>
>
> **M2: Any way to have proxy measure for assumptions in practice ?**
>
> Thanks for such insightful comments! Condition i, data diversity, is a assumption regarding to the unembedding mapping $\mathbf{g}$, this is mild assumption, since requiring generate difference vectors $\mathbf{g}_i- \mathbf{g}_0$ that are linearly dependent has measure zero, as pointed out by Roeder et al (Roeder, Geoffrey, Luke Metz, and Durk Kingma. "On linear identifiability of learned representations." ICML, 2021.). Even with this claim, we agree that if there is a reasonable proxy measure for this would be more better.
>
>
> To provide a practical diagnostic, we implemented a greedy LU-pivoting procedure that searches for a maximally diverse subset of output tokens from a candidate pool (10k tokens). Unlike random outputs—which are indeed highly correlated—we consistently find a subset whose output-difference matrix is numerically full-rank across Pythia-70m, 1.4B, and 2.8B models. The singular-value spectra (Appendix Sec. K) show smooth decay with no collapse until the final few dimensions, demonstrating that modern LLMs contain sufficiently many diverse output directions for the assumption to hold approximately.
>
> To address the concerns about the strength of the informational-sufficiency assumption, we also performed a diagnostic based on conditional entropy across the 27 concepts and three Pythia model scales (Appendix Sec. K) . The results show a clear monotonic trend: while the smallest model (70M) yields moderate entropies, the larger models (1.4b and 2.8b) consistently exhibit low conditional entropy (mostly below 0.15 bits), indicating that the representation nearly determines the target concept. This provides direct empirical support that, at model scales relevant to our theory, the informational-sufficiency (approximate invertibility) assumption is well satisfied in practice.
>
> **M3: It would be helpful to operationalise this with a small synthetic to...**
>
> **R3:** We appreciate the suggestion to complement Appendix C with a toy synthetic example.
>
> Conceptually, such a study would require us to explicitly construct two generative worlds: one where the representation is a linear mixture of joint posteriors, and one where it is a mixture of component-wise posteriors, together with known mixing matrices. This joint and component-wise different allow us to construct the following experiment. For example, with 5 binary latent variables, prior work requires a representation dimension of $2^5$ (note that the mixing matrix is invertible as in the previous work), whereas our result allows $2 \times 5$ dimensions, corresponding to mixtures over component-wise posteriors. We could therefore train a model to learn $2 \times 5$ dimensional representations and evaluate how well linear probing recovers each latent concepts. The probe achieves a classification accuracy of $0.923$ (std: 0.021). This demonstrates that $2 \times 5$ dimensional representations are already sufficient for linear classification, and that $2^5$ dimensional representations required by the previous work are unnecessary, thereby empirically supporting our theoretical result in Theorem 2.1. More details can be found in Sec. M.

---

> ### Author Response · Authors · 2025-11-26
>
> **M4. Evaluation set for Pearson correlation is quite narrow. The counterfactual corpus uses twenty seven pairs drawn from analogy like transformations. This is a good start but it is narrow.**
>
> Thank you for pointing this out. We fully agree that evaluating concept requires carefully constructed counterfactual pairs, and that collecting such data is inherently challenging, as noted in the draft.
>
> To strengthen our evaluation, we went beyond the original 27 pairs and constructed an additional set of 23 counterfactual concept pairs, covering broader linguistic phenomena, such as sentiment, toxicity polarity, factuality/truthfulness shifts  (see Appendix P for full details and statistics). Importantly, the performance gaps among methods remain consistent across the expanded 50-pair benchmark, demonstrating that our findings are not artifacts of the original 27 pairs. This additional evaluation increases both the semantic breadth and robustness of our results. We will update additional results (on Gemma and Qwen) and integrate them with the existing ones as we refine the submission, given the limited response window, and appreciate your understanding.
>
> **m: Define concept precisely and early, Figure 2 grids and Figure 3 bars,Reporting variability in Figures 3 to 5**
>
> **R---m:** Thanks for the suggestions regarding an earlier and clearer definition of “concept,” the presentation of grids in Figure 2 and bar plots in Figure 3, and reporting variability in Figures 3–5. We agree with these points and have updated the draft accordingly.
>
> For Figures 3, the results are obtained from multiple runs, and we have now included the detailed statistics in Sec. Q for completeness.

---

> > ### Comment · Reviewer_tZwq · 2025-11-26
> > **Answer**
> >
> > I have read the other reviews and the authors' detailed responses. I maintain my score of 8. I believe this paper is a solid, refreshing addition to the literature that offers a much-needed theoretical grounding in a field often dominated by not so convincing heuristic.
> >
> > On the Definition of Concepts, regarding the authors' updated clarifications: I am slightly wary of the strict equating of "latent factors organizing the data manifold" with "human-interpretable concepts." While it is true that for text, humans generally understand the semantic factors, it is not theoretically guaranteed that the true latent generative factors of a high-dimensional manifold map 1-to-1 to **"human interpretable"** language concepts. A latent factor could be a complex statistical dependency that organizes the data but has **no simple human label**. I am curious what the authors thinkgs about that ?
> >
> > Ok, because I think this is a good paper, I will try to answer to some of my fellow reviewers major points:
> > - On Experiments and Baselines: I agree with the other reviewers that the experimental suite is not as exhaustive as established methods, but I argue this is expected for a paper introducing a substantially new theoretical paradigm. The approach is novel, and demanding SOTA-beating performance or massive scale at this stage stifles innovation. The additional experiments provided in the rebuttal are sufficient proof of concept for me.
> > - On the exp Function (Response to Reviewer 57nN): Regarding the criticism of replacing exp with surrogates like SoftPlus: I do not view this as a flaw. In energy-based modelling and probabilistic deep learning, it is standard practice to parameterize energy/probability using stable surrogates. The exact exponential form is less critical than the functional behavior of enforcing sparsity in the probability domain.
> > - On Metrics (Response to Reviewers 57nN & aSXj): I agree that MSE is a weak metric for faithfulness. However, the authors do not rely solely on it.
> > - The Hungarian algorithm check (clarified in response to Reviewer aSXj) is a nice way to handle the permutation indeterminacy when calculating correlations. This was a good request from the reviewer and I think this clarify the result.
> >
> >
> > Overall, the paper provides a principled route to concept extraction that challenges current assumptions (like where to place sparsity). I believe it is worthy of acceptance.

---

> > > ### Author Response · Authors · 2025-11-27
> > >
> > > Dear Reviewer tZwq
> > >
> > > Thank you very much for your exceptionally positive assessment of our work. Beyond reviewing our submission, you even helped us respond to other reviewers’ concerns. This level of engagement is, for us, already the greatest form of encouragement.
> > >
> > >
> > > -----
> > >
> > > Regarding your question about how we should understand human-interpretable concepts: this is indeed an excellent question. Since the moment we first began engaging with work on LLM interpretability, we have been asking ourselves the same thing. What exactly constitutes a human-interpretable concept? How should it be understood, and how might it be formally defined?
> > >
> > > So far, our current thinking is as follows.
> > >
> > > ----
> > >
> > > We approach human-interpretable concepts from a latent-variable–model perspective. Suppose there exists a true latent generative model that produces the observed text $\mathbf{x}$, with underlying latent factors denoted by $\mathbf{n}$. These factors may interact in complex and nontrivial ways. Conceptually, we think of the elements of $\mathbf{n}$ as *essential and inseparable units* of the generative process, analogous (informally) to ``elementary particles'' in a physical system. In this idealized formulation, the generative process may be expressed as:
> > >
> > > $\mathbf{n} \rightarrow \mathbf{x}.$
> > >
> > >
> > > Based on this view, our current understanding is that human-interpretable concepts, denoted by $\mathbf{z}$, may correspond to a more *coarse-grained, intermediate level* between the raw generative factors $\mathbf{n}$ and the observations $\mathbf{x}$:
> > >
> > >
> > > $\mathbf{n} \rightarrow \mathbf{z} \rightarrow \mathbf{x}.$
> > >
> > >
> > > Here, each concept $z\_i$ may aggregate a *set* of underlying latent units from $\mathbf{n}$ (And, we fully agree that there might not be a strict 1-to-1 mapping!). The human concept “emotion” (e.g., “happiness” or “anger”) may bundle together multiple micro-factors such as physiological arousal, facial-expression cues, voice pitch patterns, situational context, and so on.These correspond to a set of latent elements $\{ n_j \}$, yet collectively manifest as a single human-perceived concept $z_i$.
> > >
> > > From a technical perspective, recovering all individual latent factors in $\mathbf{n}$ from $\mathbf{x}$, up to element-wise transformations, is in general impossible, consistent with well-known latent-variable identifiability limits. However, recovering
> > > the *coarse-grained* variables $\mathbf{z}$, each corresponding to a grouped set of latent micro-factors, might be a strictly less ambitious and potentially more attainable goal. Since the internal dependencies among the $n_j$ within each group do not need to be fully disentangled, $z_i$ may be more recoverable than the full fine-grained latent structure.
> > >
> > > Moreover, for many interpretability or representation-learning applications, recovering the concept-level variables $\mathbf{z}$ might be already sufficient and meaningful, even if the underlying micro-structure $\mathbf{n}$ remains inaccessible.
> > >
> > > In summary, while recovering all of $\mathbf{n}$ would be ideal yet often infeasible, recovering the concept-level coarse-grained factors $\mathbf{z}$ provides a more practical and theoretically plausible target. (There is no free lunch!)
> > >
> > > ---
> > >
> > > Best,

---

### Official Review · Reviewer_7acV · 2025-11-06

**Soundness:** 3
**Presentation:** 2
**Contribution:** 2
**Rating:** 4
**Confidence:** 3

**Summary:**

The authors suggest ConCA, which uses a probabilistic framing of how latent features can interact and be expressed in a contextful decoder language model, and unmixes the latent concepts by recoring the log-posterior of each concept.

**Strengths:**

I like the framing of this paper. It starts with a thoughtful feature-latent space definition, which allows arbitrary interactions between features, and then argues for why the linear process might be true in this case. This is a  stronger case than assuming that concepts are linearly encoded a priori.

**Weaknesses:**

W1 I do not feel that this paper offers a significant amount in terms of either contributing something to how we understand that models work, or contributing a very useful tool that will be widely used by interpretability. I feel that ConCA is well-theoretically motivated, but in practice I’m not sure what to take from it, or if there is a reason to switch to using it instead of other dictionary learning methods.

W2 The empirical results do not have too many widely-applicable takeaways. It would have been nice to see some more empirical results that validate the sentiment that you express in the last sentence of section 4 (line 471). For example, in a toy identifiability setting with a lot of interaction between latent features, is this unmixing better than something like an SAE? What kinds of interactions lead to features not getting pulled out by SAEs, but getting successfully unmixed by ConCA?

**Questions:**

Are the concepts pulled out by ConCA more easy to understand than other methods?

(no need to respond to small suggestions below)

(minor) It would be helpful to have slightly more developed captions, what each figure is showing and what we should take away from

(minor) Many of the later figures are pretty color-based, it might be worth seeing if there are ways to make them more readable.

(v. minor) there are a few typos in the paper, a few are verb inflection typos, perhaps it’s worth a read-through looking just at that

---

> ### Author Response · Authors · 2025-11-26
>
> **Q1: I feel that ConCA is well-theoretically motivated, but in practice I’m not sure what to take from it**
>
> **R1:** We thank the reviewer for highlighting the value of ConCA as a theory-driven approach. Indeed, theory is the central thread of our entire work.
>
> Existing SAEs are motivated by two heuristic hypotheses—the linear representation hypothesis and the superposition hypothesis. While these hypotheses provide useful **coarse-level** intuition, they are not sufficiently fundamental to uniquely guide the design of a concept extraction method. For example, the superposition hypothesis implies that sparsity may help disentangle features, but where should sparsity be applied? On the encoder output or on a transformed space? (if yes, which transformed space?) These questions may not be rigorously answered within the two hypotheses, mainly because these hypotheses do not describe the internal structure of LLM representations at a sufficiently **deep level**.
>
> Motivated by this gap, we seek a more principled theoretical foundation. This leads to our key result—Theorem 2.1, which shows that the internal representation of an, is well-approximated by a linear mixture of the marginal log-posteriors of the latent concepts:
>
> Building on this theoretical understanding, we **revisit** and **challenge** several conventional SAE design choices **in practice**:
>
> 1) *Challenging the Nonlinear Activation Function (typically, ReLU):* Guided by our theory, the objective of concept extraction is linear unmixing. Under this objective, the nonlinear activation functions commonly used in SAE encoders (such as ReLU) are not theoretically justified and may introduce distortions that hinder accurate unmixing. ConCA therefore adopts an architecture that avoids unnecessary nonlinearities, yielding both more flexible and more faithfully aligned with the underlying theory.
>
> 2) *Challenging the Conventional Placement of Sparsity:* Our goal is to recover the component-wise log-posteriors $\log p(z_{i}|\mathbf{x})$. When $\log p(z_{i}|\mathbf{x})=0$, it means the probability $p(z_{i}|\mathbf{x})=e^0=1$ (the concept is fully active). Therefore, imposing sparsity directly on the feature space may be reasonable. Our theory mandates that sparsity must be enforced on the **exponentially transformed** feature space.
>
> 3) *Challenging the Contested Evaluation Metric of SAEs:* SAE evaluation often relies on heuristic metrics, which lack theoretical principles and are often subject to debate11. ConCA introduces a principled, theory-backed evaluation metric: based on Corollary 3.1, we demonstrate that linear probing performance can serve as a theoretically consistent reference (or upper bound proxy) to quantify the faithfulness of the extracted features to the true conceptual log-posterior $\log p(z_i|\mathbf{x})$
>
> In summary, ConCA is not merely a theoretically more elegant alternative, it represents a paradigm shift that rethinks the traditional SAE framework based on theoretical grounding. Our empirical results (higher MPC and stronger downstream generalization) validate that these theory-driven design choices lead to more faithful, stable, and transferable interpretable features. Thus, ConCA constitutes a significant contribution to the interpretability toolkit.

---

> ### Author Response · Authors · 2025-11-26
>
> **Q2: More empirical results that validate the sentiment that you express in the last sentence of section 4 (line 471). In a toy identifiability setting with a lot of interaction between latent features, is this unmixing better than something like an SAE? What kinds of interactions lead to features not getting pulled out by SAEs, but getting successfully unmixed by ConCA?**
>
> **R2:** Thanks for your comments.
>
> Before presenting the additional experimental results you requested, we would like to clarify two points:
>
>
> 1. Clarification of the Sentence in Question. The sentence in Section 4 (line 471):
>
> “The improvements above are likely attributable to ConCA’s principled framework, grounded in Theorem 2.1, which …”
>
> serves as a potential explanation for performance gains that have already been empirically observed. Specifically, ConCA’s superiority has been consistently demonstrated across a large-scale evaluation involving 113 diverse few-shot learning datasets and 8 challenging OOD benchmarks. This extensive empirical evidence firmly establishes the broad applicability and robustness of ConCA’s learned features.
>
> 2. Scope of Contribution and Latent Interaction. The reviewer’s question assumes that the performance gap between ConCA and SAEs is explained by the presence or complexity of latent interactions (Please let us known, if we misunderstand your comments). However, this assumption falls outside the scope of our theoretical claims. Our identifiability result (Theorem 2.1) holds regardless of whether latent factors exhibit simple, strong, or arbitrary dependencies. Consequently, our work does not attribute ConCA’s advantages to handling particular types of latent interactions. Instead, the empirical improvements instead arise from the principled target, i.e., avoiding unnecessary nonlinearities and enforcing sparsity on exponentially transformed feature space, as highlighted in the response to your question 1, which SAEs fundamentally lack.
>
> 3. We conducted this toy setup, the five latent variables form an Erdős–Rényi DAG with substantial dependencies (expected 10 edges among 5 variables), ensuring strong interaction and correlation among latent concepts. To simulate a nonlinear mixture process, we then convert the latent variable samples into one-hot format and randomly apply a permutation matrix to the one-hot encoding, generating one-hot observed samples. To simulate next-token prediction, we randomly mask a part of the binary observed data, e.g.,, ${x}_i$, and predict it by use the remaining portion $\mathbf{x}_{\setminus i}$. Under this context, the Pearson correlation between logits (obtained by linear-probe logits) and SAE features is 0.635 (std 0.046). In contrast, ConCA achieves a substantially higher correlation of 0.813 (std 0.031). Please see further discussion in Sec. M.

---

> ### Author Response · Authors · 2025-11-26
>
> **Q3: Are the concepts pulled out by ConCA more easy to understand than other methods?**
>
> We would like to highlight the following advantages of ConCA, comapred with SAEs.
>
> Theoretical Clarity and Target Precision: ConCA features are designed to recover the log-posterior probability of individual latent concepts. This is a clearly defined probabilistic quantity that directly measures how strongly input activates concept. In contrast, SAE features are abstract basis vectors in the neural representation space, lacking a clear theoretical mapping to interpretable concepts. ConCA thus provides features with explicit semantic meaning and well-defined theoretical bounds.
>
> Empirical Faithfulness and Alignment: We quantify alignment between ConCA features and a supervised estimate of log-posterior of concepts given input context (derived via linear probing logits) using the Pearson Correlation Coefficient (MPC).  ConCA consistently achieves higher MPC than SAEs, demonstrating that its features more faithfully approximate the model’s internal concept probabilities, enhancing interpretability.
>
> Stronger Monosemanticity: ConCA’s principled design, particularly the correct application of sparsity in the probability domain, produces features with clearer, single-meaning semantics.
>
> Stability Evidence: Counterfactual analyses using the Rank-based Fraction of Significant Features (Figure 4) show that ConCA features exhibit minimal spurious variation when only a single concept is changed, indicating reduced interference.
>
>
> Summary: By combining a theoretically precise target with principled sparsity and strong empirical alignment, ConCA extracts features that are not only faithful to the underlying model but also more interpretable and stable than those produced by SAEs.

---

### Author Response · Authors · 2025-11-26
**General Response**

We would like to sincerely thank all reviewers for taking the time to carefully evaluate our work.

We truly appreciate the thoughtful comments and constructive suggestions, many of which have substantially helped improve the clarity and quality of the paper.

Some of the additional experiments required significant computational time, and we apologize for the delay in submitting our response. We are grateful for your patience and understanding.

---

### Meta-Review · Area_Chair_VNN9 · 2026-01-13

**Summary:**

Concept Component Analysis follows a long line of prior work on identifying interpretable features in neural networks, and more recently concept learning in LLMs via SAEs. The main theorem shows that under assumptions the model can be approximated as a linear mixture of stacked log posteriors of latent concepts.

Some reviewers rightfully had some difficulties pinpointing the significance of the theory, especially given the proliferation of recent theoretical work on this problem that is cited but not carefully discussed. The response does not seem to address this concern, and the paper reads as if very little prior work has been done. The authors should do a better job of situating and contextualizing their work within prior work on both theory and empirics, much of which is cited but not discussed adequately or in the right places.

**Reviewer Concerns:**

- 7acV Significance: **No**
- 7acV Empirical takeaways: **No**
- tZwq Mixing matrix: Yes
- tZwq Proxy measure for assumptions: Yes
- tZwq Identifiability: Yes
- 57nN exp function is too numerically unstable: **No**
- 57nN MSE is a weak metric: **No**
- 57nN Footnote 2: Yes
- 57nN Error bars in Figure 5: Yes
- aSXj Main Fig 2: Yes
- aSXj Rationale of mean pearson correlation: Yes
- aSXj Theory gap: **No**
- aSXj Linear autoencoder: Yes
- aSXj Relation to word embeddings: **No**

**Reviewer Scores:**

7acV 4->4
tZwq 8->8
57nN 4->6
aSXj 4->4

---

### Decision · Program_Chairs · 2026-01-26

Reject